# Cell-specific and divergent roles of the CD40L-CD40 axis in atherosclerotic vascular disease

Michael Lacy [1,2,13], Christina Bürger[1,13], Annelie Shami[3,13], Maiwand Ahmadsei[1,2,4], Holger Winkels[1,5], Katrin Nitz[1,2], Claudia M. van Tiel[6], Tom T. P. Seijkens[6], Pascal J. H. Kusters[6], Ela Karshovka[1], Koen H. M. Prange[6], Yuting Wu[1], Sanne L. N. Brouns[7], Sigrid Unterlugauer[1], Marijke J. E. Kuijpers[7], Myrthe E. Reiche[6], Sabine Steffens[1,2], Andreas Edsfeldt[3,8,9], Remco T. A. Megens [1], Johan W. M. Heemskerk [7], Isabel Goncalves[3,8], Christian Weber[1,2,7,10], Norbert Gerdes [1,11✉], Dorothee Atzler [1,2,12✉] & Esther Lutgens [1,2,6✉]

Atherosclerosis is a major underlying cause of cardiovascular disease. Previous studies showed that inhibition of the co-stimulatory CD40 ligand (CD40L)-CD40 signaling axis profoundly attenuates atherosclerosis. As CD40L exerts multiple functions depending on the cell-cell interactions involved, we sought to investigate the function of the most relevant CD40L-expressing cell types in atherosclerosis: T cells and platelets. Atherosclerosis-prone mice with a CD40L-deficiency in CD4$^+$ T cells display impaired Th1 polarization, as reflected by reduced interferon-γ production, and smaller atherosclerotic plaques containing fewer T-cells, smaller necrotic cores, an increased number of smooth muscle cells and thicker fibrous caps. Mice with a corresponding CD40-deficiency in CD11c$^+$ dendritic cells phenocopy these findings, suggesting that the T cell-dendritic cell CD40L-CD40 axis is crucial in atherogenesis. Accordingly, sCD40L/sCD40 and interferon-γ concentrations in carotid plaques and plasma are positively correlated in patients with cerebrovascular disease. Platelet-specific deficiency of CD40L does not affect atherogenesis but ameliorates atherothrombosis. Our results establish divergent and cell-specific roles of CD40L-CD40 in atherosclerosis, which has implications for therapeutic strategies targeting this pathway.

[1] Institute for Cardiovascular Prevention (IPEK), Ludwig-Maximilians-Universität, Munich, Germany. [2] German Centre for Cardiovascular Research (DZHK), partner site Munich Heart Alliance, Munich, Germany. [3] Department of Clinical Sciences Malmö, Lund University, Clinical Research Center, Malmö, Sweden. [4] Klinik und Poliklinik für Innere Medizin I, Klinikum rechts der Isar, Technical University of Munich, Munich, Germany. [5] Department III for Internal Medicine, University Hospital Cologne, Cologne, Germany. [6] Department of Medical Biochemistry, Amsterdam Cardiovascular Sciences (ACS), Amsterdam University Medical Centers, University of Amsterdam, Amsterdam, The Netherlands. [7] Department of Biochemistry, Cardiovascular Research Institute Maastricht (CARIM), Maastricht University, Maastricht, The Netherlands. [8] Department of Cardiology, Skåne University Hospital, Lund University, Malmö, Sweden. [9] Wallenberg Centre for Molecular Medicine, Lund University, Lund, Sweden. [10] Munich Cluster for Systems Neurology (SyNergy), Munich, Germany. [11] Division of Cardiology, Pulmonology and Vascular Medicine, Medical Faculty, University Hospital Düsseldorf, Düsseldorf, Germany. [12] Walther-Straub-Institute of Pharmacology and Toxicology, Ludwig-Maximilians-Universität, Munich, Germany. [13] These authors contributed equally: Michael Lacy, Christina Bürger, Annelie Shami. ✉email: Norbert.Gerdes@med.uni-duesseldorf.de; dorothee.atzler@med.uni-muenchen.de; Esther.Lutgens@med.uni-muenchen.de

A therosclerotic cardiovascular disease (CVD), including myocardial infarction, stroke and peripheral arterial occlusive disease, remains the leading cause of morbidity and mortality worldwide[1]. In conjunction with hyperlipidemia, inflammation is an important driver of atherosclerosis[2,3]. The Canakinumab (anti-interleukin (IL)1ß) Anti-Inflammatory Thrombosis Outcome Study trial[4], COLchicine Cardiovascular Outcomes Trial[5] and LoDoCo2[6] trial results demonstrate that immune regulation on top of optimal lipid management reduces CV events in humans and, more importantly, emphasize the immune system's critical role in atherosclerosis.

Besides the IL1ß/inflammasome pathway, co-stimulatory immune checkpoints play a pivotal role in atherosclerosis[7]. CD40L, especially, has been shown to be a critical and powerful modulator of immune responses in atherosclerosis[8]. In humans, CD40L expression peaks in late stage atherosclerotic plaques, and its expression is associated with features of plaque vulnerability[8]. In many studies[9–11], but not all[12], increased serum levels of the soluble form of CD40L (sCD40L) are associated with hypercholesterolemia, stroke, diabetes, and acute coronary syndrome (ACS) and predict recurrent CVD.

Genetic deficiency or antibody-mediated inhibition of CD40L is highly effective in reducing atherosclerosis, even in established disease conditions, by generating plaques that are rich in collagen and contain a limited amount of immune cells resembling the murine equivalent of a stable, clinically safe, plaque[13–16]. Targeting CD40L is thus a favorable therapeutic strategy to combat atherosclerotic CVD. However, translational studies applying CD40L inhibition in chronic inflammatory diseases revealed that while global inhibition may be effective, it may not be a viable therapy. Inhibition of CD40L resulted in unstable thrombi, which are embolized easily and cause thrombo-embolic events[17,18]. To reduce these severe effects, targeted approaches are necessary; however, the cell source requirements for CD40L in atherosclerosis remain unclear.

A multitude of immune cells involved in the pathogenesis of atherosclerosis express CD40L including various lymphocyte and myeloid subpopulations, but also platelets, endothelial cells and vascular smooth muscle cells (VSMCs)[19,20]. Even though CD40L expression may be broad, T cells and platelets are the primary expressers of this co-stimulatory molecule[8,21]. In T cells, CD40L acts together with CD40 expressed on antigen presenting cells as a secondary signal to amplify initial T-cell receptor signaling[22], and CD40L helps induce T-cell activation and immunoglobulin-isotype switching. Although CD40 is the most common receptor for CD40L in immune cells[23], CD40L can also bind to the integrin complexes $\alpha_M\beta$[24] and $\alpha5\beta1$[25]. Platelet CD40L, on the other hand, mainly signals through the integrin $\alpha$IIb$\beta$3 leading to platelet aggregation and endothelial activation[26,27].

Until now, it is unclear which CD40L expressing cell type(s) are responsible for the different aspects of atherogenesis. In the past, CD40L[−/−] bone marrow chimeras did not show any effect on atherosclerosis, suggesting that CD40L-expressing non-hematopoietic cells, i.e., endothelial cells or VSMCs, drive atherosclerosis[28,29]. On the other hand, transfer of thrombin-activated CD40L-deficient platelets ameliorated platelet-induced aggravation of atherosclerosis by preventing platelet-leukocyte aggregate formation as well as leukocyte recruitment, suggesting a major role for platelet CD40L in atherosclerosis[27]. However, both approaches are rather unspecific: in a bone marrow transplantation, all hematopoietic lineages are transplanted, and the sublethal irradiation, required for the procedure, induces damage to the bone marrow stroma, which may impact mature immune cell function. Transfer of thrombin-activated CD40L[−/−] platelets, only reflects the effects of platelet CD40L after strong activation,

and although informative, does not delineate the role of platelet CD40L in atherogenesis.

In this work, we therefore investigated the contribution of T cell- and platelet-specific CD40L in atherosclerosis using conditional gene-deficient models. Here, we show that T cell-specific CD40L deficiency reduces atherosclerotic plaque development through impaired T helper (Th) 1 polarization, which in turn leads to more stable lesions. Combining flow cytometric, transcriptomic, and in vitro antigen-specific T cell cytokine analyses, we show that the T cell—dendritic cell (DC) CD40L-CD40 dyad is critical for interferon-γ production. Deficiency of platelet CD40L, on the other hand, does not affect atherogenesis. However, using an atherothrombosis model with activated platelets, deficiency of platelet CD40L reduces neointima formation. These findings provide insight into the cell-specific roles of the CD40L-CD40 dyad in atherosclerosis, which may be valuable for developing targeted therapies.

## Results

**Baseline characteristics of T cell-specific (Cd40l^fl/fl/Cd4Cre) and platelet-specific (Cd40l^fl/fl/Pf4Cre) CD40L deficient mice as well as dendritic cell specific (Cd40^fl/fl/Cd11cCre) CD40 deficient mice.** To explore the effect of CD40L in T cells and platelets in atherosclerosis, we generated mice with a T cell-specific (referred to as Cd40l^fl/fl/Cd4Cre^tg) or platelet-specific deletion of CD40L (referred to as Cd40l^fl/fl/Pf4Cre^tg) and backcrossed them to Apoe^−/− mice to induce atherosclerosis (Supplementary Fig. 1a). For further analysis of the CD40L-CD40 T cell—DC axis in atherosclerosis, we also generated mice deficient for CD40 in CD11c^+ DCs (referred to as Cd40 ^fl/fl/Cd11cCre^tg), which were backcrossed to Apoe^−/− mice (Supplementary Fig. 2a). Cell-type specific deletions of all CD40(L) models were confirmed (Supplementary Figs. 1b-d and 2b-c). Breeding pairs were heterozygous for the Cre transgene to obtain both Cre^tg animals and Cre^wt animals. The resulting Cre^wt mice (i.e., Cd40l^fl/fl/Cd4Cre^wt, Cd40l^fl/fl/Pf4Cre^wt, Cd40 ^fl/fl/Cd11cCre^wt, all on an Apoe^−/− background) served as respective littermate controls.

With the exception of a slight increase in peripheral white blood cells including lymphocytes, no major differences were observed in body weight, lipid levels, or basic hematologic parameters (Supplementary Table 1) between cell-specific knockout models and their littermate controls.

**T cell-specific CD40L decreases atherosclerosis.** Male Cd40l^fl/fl/Cd4Cre, Cd40l^fl/fl/Pf4Cre, and Cd40 ^fl/fl/Cd11cCre mice consumed a normal chow diet for 28 weeks and all developed advanced atherosclerotic plaques in the aortic root. In the absence of T cell CD40L, plaque area significantly decreased by 28% (Fig. 1a). Plaques from Cd40l^fl/fl/Cd4Cre^tg mice were not only smaller, but also contained fewer CD4^+ T cells (Fig. 1b). The percentage of plaque macrophages (Mac3^+) had increased in Cd40l^fl/fl/Cd4Cre^tg mice (Fig. 1c) suggesting Cd40l^fl/fl/Cd4Cre^tg plaques were less advanced compared to their littermate controls considering these were late stage atherosclerotic plaques. Indeed, the reduction in atherosclerotic plaque in the aortic root of Cd40l^fl/fl/Cd4Cre^tg mice was associated with a shift from fibrous cap atheromas (FCAs) toward earlier plaque phenotypes according to a Virmani classification[30] (Fig. 1d). Although there was a stark decrease in FCAs in the absence of T cell CD40L, analysis of the minimum fibrous cap thickness in advanced plaques revealed that Cd40l^fl/fl/Cd4Cre^tg mice had significantly thicker fibrous caps (Fig. 1e) and a higher smooth muscle cell (SMC) content (Fig. 1f) compared to their wild type littermates. Therefore, absence of CD40L appeared to slow plaque progression as well as stabilize late-stage atherosclerotic plaques. In addition to the redistribution of collagen

content, the reduction in necrotic cores (Fig. 1g) observed in *Cd40l^fl/fl/Cd4Cre^tg* mice further solidified the shift toward a more stable and clinically favorable plaque phenotype.

To check for underlying immunological mechanisms explaining this plaque phenotype, we first analyzed the expression of several pro-inflammatory and anti-inflammatory cytokines together with T-cell markers in the descending aorta of *Cd40l^fl/fl/Cd4Cre^tg* mice and their wild type littermates. Accordingly, transcripts of several pro-inflammatory cytokines including *interleukin (IL)-6* and *interferon*

*(IFN)-γ* were decreased in the absence of T cell CD40L while anti-inflammatory cytokines such as *transforming growth factor (TGF)-ß* and *IL-10* remained unchanged (Fig. 1h). A similar decrease in atherosclerosis with a subsequent decrease in plaque T cells and aortic *IFN-γ* expression as well as an increase in plaque SMCs was observed in our DC-specific CD40 knockout mice (Supplementary Fig. 3a–d), suggesting that the CD40L-CD40 T cell—DC axis plays a critical role in polarizing T cell-mediated inflammation in atherosclerosis.

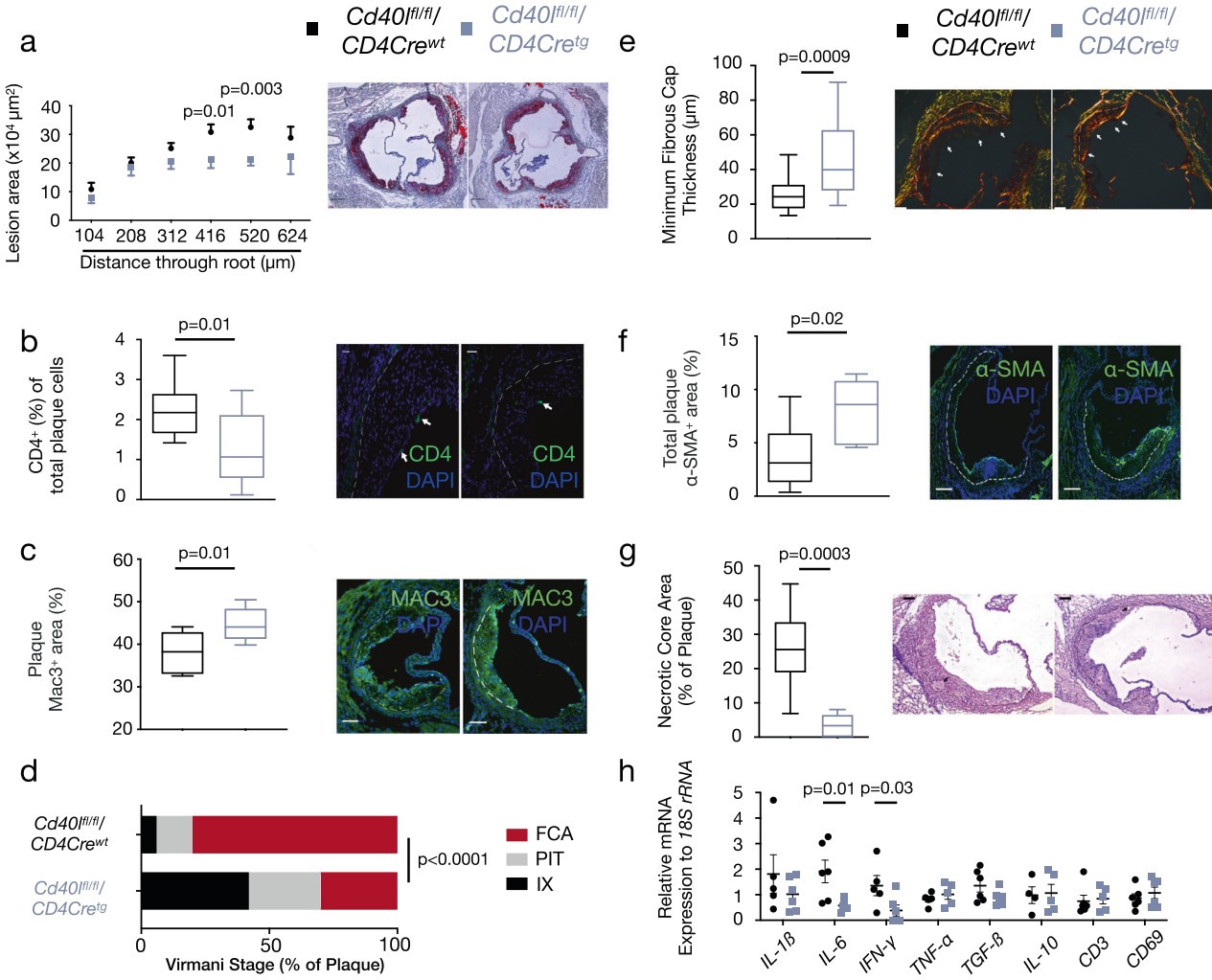

**Fig. 1 Lack of T cell CD40L reduces plaque burden with less advanced, more stable plaques. a** Atherosclerotic plaque area in cross-sections at indicated positions of the aortic root from *Cd40l^fl/fl/Cd4Cre* mice [*n*: WT = 17, TG = 16] with representative Oil Red O-stained photomicrographs (scale bar: 200 μm). **b**, **c** Immunofluorescent staining assessing plaque phenotypes in cross-sections of the aortic root of mice with quantifications (left) and representative photomicrographs (right). Arrows indicate stained cells and the dashed line separates the atherosclerotic lesion. **b** CD4+ T cells (green) along with nuclear DAPI staining (blue) [*n* = 10] and (**c**) Mac3+ area (green) along with nuclear DAPI staining (blue) [*n* = 7] (scale bar: 100 μm). **d** Chi-square distribution analysis of morphological plaque phenotypes describing initial xanthoma (IX), pathological intimal thickening (PIT), and fibrous cap atheroma (FCA) phenotypes [*n*: WT = 17 animals/51 lesions, TG = 13/39 lesions]. **e–g** Multi-parameter analysis of plaque stability assessing: **e** Fibrous cap thickness [*n*: WT = 20, TG = 14] using a Sirius Red stain (polarized light), (**f**) Alpha-smooth muscle actin+ (α-SMA+) area [*n*: WT = 9, TG = 5] (green, scale bar: 75 μm), and (**g**) Necrotic core content [*n* = 8] using hematoxylin and Eosin stain all in aortic root sections with quantifications (left) and representative photomicrographs (right). **h** Gene expression analysis using qPCR to evaluate various mRNA cytokine expression profiles in the descending aorta [*n*: WT = 6: CD3, TGF-ß, IL-6, CD69; 5: IL-1ß, 4: IFN- γ, IL-10, TNF- α; TG = 7: TGF-ß; 6: IL-6, CD69; 5: IL-1ß, IL-10, IFN- γ, CD3; 4: TNF- α. Data are represented as either mean ± s.e.m. **a**, **h** or boxplots (**b**, **c**, **e–g**). In the boxplots, whiskers above and below the boxes indicate the 10th and 90th percentile while the boundaries of the boxes represent the 25th and 75th percentile. The black line within the boxes marks the median. n refers to biologically independent animals. Data were analyzed by either a two-tailed unpaired Student's *t* test (**a–c**, **f**), two-tailed Mann–Whitney test (**e**, **g**, **h**), or Chi-square test (**d**). Interleukin-1 beta IL-1ß, Interleukin-6 IL-6, Interferon-gamma IFN-γ, Tumor necrosis factor-alpha TNF-α, Transforming growth factor-beta TGB-ß, Interleukin-10 IL-10, Cluster of differentiation 3 CD3. Source data are provided as a Source Data file.

**sCD40L and sCD40 levels correlate with IFN-γ in human plasma and atherosclerotic plaques**. To investigate whether the CD40(L)-IFN-γ axis, observed in our in vivo mice models, is also present in patients, sCD40L and sCD40 levels were measured in atherosclerotic carotid plaques and plasma samples from 185 patients from the Carotid Plaque Imaging Project (CPIP) cohort (Supplementary Table 2). Within the advanced carotid atherosclerotic plaques, both sCD40L ($r = 0.264$, $p = 0.000321$) and sCD40 ($r = 0.211$, $p = 0.004$) were positively correlated (Spearman correlation) with levels of IFN-γ in the plaque. Likewise, plasma sCD40L ($r = 0.276$, $p = 0.000113$) and sCD40 ($r = 0.186$, $p = 0.010$) concentrations also correlated (Spearman correlation) with circulating IFN-γ levels. These data suggest that, similarly to our hyperlipidemic mice, the CD40L-CD40 pathway also initiates a Th1-associated IFN-γ response in human atherosclerosis.

**Platelet-specific CD40L is only involved in atherothrombosis**. Platelet-specific CD40L deficiency, on the other hand, did not affect atherosclerotic plaque burden (Fig. 2a) or plaque phenotype. Plaques of $Cd40l^{fl/fl}/Pf4Cre^{tg}$ were of similar size and showed no alterations in the amount of T cells (Fig. 2b), macrophages (Fig. 2c) or SMCs (Fig. 2d). As platelet CD40L is known for its role in thrombosis, we performed whole blood perfusion microspot experiments using $Cd40l^{fl/fl}/Pf4Cre^{wt}$ and $Cd40l^{fl/fl}/Pf4Cre^{tg}$ platelets. Deficiency of platelet CD40L led to a 28% reduction in platelet deposition (Fig. 2e) and fibrinogen content in the formed thrombi (Fig. 2f) thereby suppressing thrombus build-up (Fig. 2g). These data suggest that platelet CD40L is more important in the pathogenesis of atherothrombosis rather than atherosclerosis. Indeed, following wire injury, thrombus- and injury-induced lesion growth in $Cd40l^{fl/fl}/Pf4Cre^{tg}$ mice was significantly reduced compared to their littermate controls (Fig. 2h).

**T cell-specific CD40L deficient mice have profound reductions in Th1 responses**. To understand the marked decrease in atherosclerosis and pro-inflammatory cytokines observed in $Cd40l^{fl/fl}/Cd4Cre^{tg}$ mice, we characterized their immune cells via flow cytometry. In the blood, we observed no significant changes in major immune cell populations (Supplementary Fig. 4a). However, as we already observed in our histological analysis, T cell-specific CD40L deficiency affected major CD4$^+$ subpopulations.

We discovered that $Cd40l^{fl/fl}/Cd4Cre^{tg}$ mice have a profound reduction in (CD4$^+$CD62L$^-$CD44$^+$) effector memory (EM) T cells in the blood (38% reduction), lymph nodes (LN, 42% reduction), and spleen (37% reduction) relative to all CD4$^+$ cells (Fig. 3a) as well as to all leukocytes (Supplementary Fig. 4b, see Supplementary Fig. 5 for gating strategy). Within this EM T-cell population, helper T cells can be distinguished by their expression of C-X-C motif chemokine receptor 3 (CXCR3) and C-C motif chemokine receptor 6 (CCR6). Th1 (CXCR3$^+$CCR6$^-$) cells were significantly reduced in the LNs and spleen of $Cd40l^{fl/fl}/Cd4Cre^{tg}$ mice while Th2 (CXCR3$^-$CCR6$^-$) and Th17 (CXCR3$^-$CCR6$^+$) cell populations remained unchanged (Fig. 3b-c). As reported above, gene expression analysis in the descending aorta reported similar findings with a 77% reduction in the Th1-associated cytokines IFN-γ. DC-specific ($Cd40^{fl/fl}/Cd11cCre^{tg}$) CD40 deficient mice phenocopied this decrease in Th1 cells in the LNs and spleen (Supplementary Fig. 3e).

The other CD4$^+$ subpopulation that was affected were the CD25$^+$FoxP3$^+$ T regulatory cells (Tregs). Tregs were reduced in the blood, LNs, and spleens of $Cd40l^{fl/fl}/Cd4Cre^{tg}$ mice relative to the total CD4$^+$ population (Fig. 3d) as well as all leukocytes (Supplementary Fig. 4c). Consistent with this finding, immunohistochemistry analysis of aortic root plaques displayed decreased

Treg presence in $Cd40l^{fl/fl}/Cd4Cre^{tg}$ mice compared to littermate controls (Supplementary Fig. 4d). Similarly, a systematic decrease in Tregs was found in $Cd40^{fl/fl}/Cd11cCre^{tg}$ mice (Supplementary Fig. 3f). Unaltered Annexin-V$^+$ thymic Tregs of $Cd40l^{fl/fl}/Cd4Cre^{tg}$ mice suggested a developmental effect in CD40L-deficient T cells rather than augmented apoptosis (Supplementary Fig. 4e). To identify differences in the functional abilities of Tregs isolated from $Cd40l^{fl/fl}/Cd4Cre^{tg}$ mice, we performed a suppression assay. Notably, we observed no difference in the suppressive capacity of Tregs isolated from $Cd40l^{fl/fl}/Cd4Cre^{tg}$ mice compared to Tregs from their littermate controls (Fig. 3e). Despite diminished Treg populations, $Cd40l^{fl/fl}/Cd4Cre^{tg}$ and $Cd40^{fl/fl}/Cd11cCre^{tg}$ mice still exhibited a reduced and more stable plaque phenotype, suggesting that the decrease in Th1 responses affect atherogenesis more strongly than the decrease in Tregs.

To further unravel the T-cell phenotype of our $Cd40l^{fl/fl}/Cd4Cre^{tg}$ mice, we performed RNA sequencing of splenic CD4$^+$ T cells from $Cd40l^{fl/fl}/Cd4Cre^{tg}$ mice and their littermate controls. In total, we discovered 400 differentially expressed genes ($p_{adj} < 0.05$) with 121 genes upregulated and 279 downregulated genes in $Cd40l^{fl/fl}/Cd4Cre^{tg}$ CD4$^+$ T cells. Interestingly, transcripts for both CXCR3 and FoxP3 were significantly downregulated in $Cd40l^{fl/fl}/Cd4Cre^{tg}$ mice matching the observed phenotypic reductions in Th1 effector cells and Tregs (Fig. 4a). Pathway analysis of differentially expressed genes revealed disruptions of Th1-associated IFN-γ signaling as well as the Treg-associated IL-2/STAT5 signaling in $Cd40l^{fl/fl}/Cd4Cre^{tg}$ mice marking the CD40L-CD40 dyad as a potentially critical pathway for the development of both T-cell subsets (Fig. 4b).

**Absence of CD40-CD40L ligation reduces IFN-γ production upon DC-T cell interactions**. Given that the reduced abundance of T-cell subpopulations may be a consequence of potential impaired reactive abilities, we investigated changes in proliferation of T cells. However, CD4$^+$ T cells stimulated with anti-CD3/CD28 beads for 72 h revealed no difference in cell expansion between $Cd40l^{fl/fl}/Cd4Cre^{tg}$ mice and their littermates (Supplementary Fig. 6a).

Although antigen-independent proliferation was unchanged, we questioned whether the absence of CD40-CD40L co-stimulation may impair specific effector T-cell responses. In a DC-OT-II T-cell co-culture, where DCs with and without CD40 were loaded with the ovalbumin (OVA) peptide and co-cultured with naïve OT-II T cells for 72 h, we examined antigen-dependent T-cell activation (Fig. 5a-b). As expected, OVA stimulation enabled DCs to activate naïve T cells while DCs stimulated with the vehicle control were unable to induce activation into effector T cells (Fig. 5c). In the absence of CD40, the ability of DCs to activate naïve T cells (Fig. 5d) was reduced. Notably, the absence of CD40 ligation resulted in a significant decrease in IFN-γ (Fig. 5e), but no change was observed in cytokines associated with other effector or regulatory T cells including IL-2, IL-10, IL-4, IL-5, IL-13, or IL-17A (Supplementary Fig. 6b-c) suggesting that the DC-T cell CD40-CD40L axis selectively inhibits Th1 responses.

**T cell-specific CD40L deficiency reduces oxLDL IgG production**. As the T cell—B cell CD40L-CD40 axis was reported to be crucial for antibody production and immunoglobulin (Ig) isotype switching[31], we investigated whether these data could be reproduced in our $Cd40l^{fl/fl}/Cd4Cre^{tg}$ model. Deficiency of T cell CD40L indeed resulted in a decrease in splenic IgD$^+$CD21$^{-/low}$ follicular B cells along with a concomitant increase in IgD$^-$CD21$^{low}$ marginal zone B cells (Supplementary Fig. 7a). Furthermore, we found a stark reduction of germinal center (GC) B cells in $Cd40l^{fl/fl}/Cd4Cre^{tg}$ mice as analyzed by co-expression of

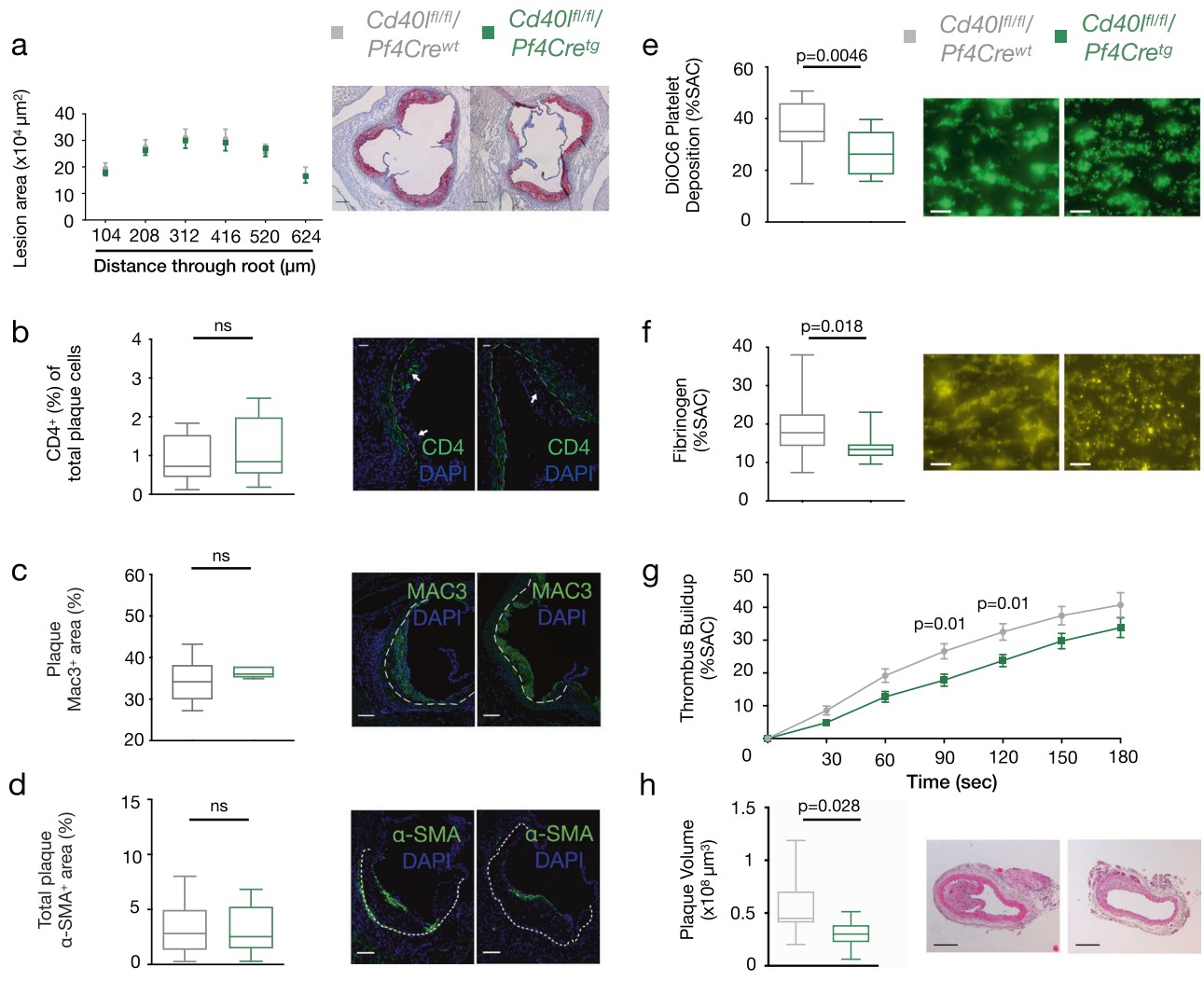

**Fig. 2 Platelet CD40L deficiency alters thrombosis formation and atherothrombosis. a** Atherosclerotic plaque area in cross-sections at indicated positions of the aortic root from *Cd40l*<sup>fl/fl</sup>*/Pf4Cre* mice [*n*: WT = 21, TG = 18] with representative Oil Red O-stained photomicrographs (scale 200 μm). **b–d** Immunofluorescent staining assessing plaque phenotypes in cross-sections of the aortic root of mice with quantifications (left) and representative photomicrographs (right). Arrows indicate stained cells and the dashed line separates the atherosclerotic lesion. **b** CD4+ T cells (green) along with nuclear DAPI staining (blue) [*n*: WT = 10, TG = 8], (**c**) Mac3+ area (green) along with nuclear DAPI staining (blue) [*n*: WT = 6, TG = 5], and (**d**) Alpha-smooth muscle actin+ (α-SMA+) area [*n*: WT = 18, TG = 15] (green, scale bar: 75 μm). **e–f** Multi-parameter assessment of thrombus formation in flowed whole blood with CD40L deficient platelets with quantifications (left) and representative photomicrographs (right, scale bar: 20 μm). Parameters evaluated after 4 min of blood flow were: **e** 3,3'-Dihexyloxacarbocyanine iodide (DiOC6) labelled platelet adhesion (Surface Area Coverage, SAC%) [*n*: WT = 19, TG = 12] and (**f**) Fluorescein isothiocyanate (FITC)-labelled fibrinogen staining of thrombi [*n*: WT = 20, TG = 12]. **g** Thrombus build-up [*n*: WT = 20, KO = 12] was analyzed using micro-spot perfusions comparing *Cd40l*<sup>fl/fl</sup>*/Pf4Cre*<sup>tg</sup> mice and their littermate controls. **h** Plaque volume [*n* = 8] was measured in the carotid arteries following wire injury with quantifications (left) and representative photomicrographs (right, scale bar: 200 μm). Data are represented as either mean ± s.e.m. (**a**, **g**) or boxplots (**b–f**, **h**). In the boxplots, whiskers above and below the boxes indicate the 10th and 90th percentile while the boundaries of the boxes represent the 25th and 75th percentile. The black line within the boxes marks the median. *n* refers to biologically independent animals. Data were analyzed by either a two-tailed unpaired Student's *t* test (**a**, **e**, **g**) or two-tailed Mann–Whitney test (**b–d**, **f**, **h**). Source data are provided as a Source Data file.

the GL-7 epitope together with peanut agglutinin (GL-7+PNA+) (Supplementary Fig. 7b-c, see Supplementary Fig. 5 for gating strategy). Accordingly, we observed a marked decrease in plasma titers of IgG2b, IgE, and IgM in hyperlipidemic *Cd40l*<sup>fl/fl</sup>*/Cd4Cre*<sup>tg</sup> mice compared to their littermate controls (Supplementary Fig. 7d). To clarify this result in the context of atherosclerosis, we also investigated the concentration of atherosclerosis-associated Igs reactive to oxidized low density lipoprotein (oxLDL) in the plasma. In line, we found a reduction in oxLDL-specific total IgG titers with a specific decrease in IgG2b levels (Supplementary

Fig. 7e). These results confirm a reduction in Ig production along with a slight impairment in Ig class switching.

## Discussion
The results of this study show that CD40L plays a divergent, cell type-specific role in atherosclerosis and highlight a crucial role for the CD40L-CD40 T cell-DC axis in atherogenesis. Deficiency of either T cell CD40L or CD40 on DCs strongly reduces lesion size, as well as plaque inflammation, resulting in a favorable stable

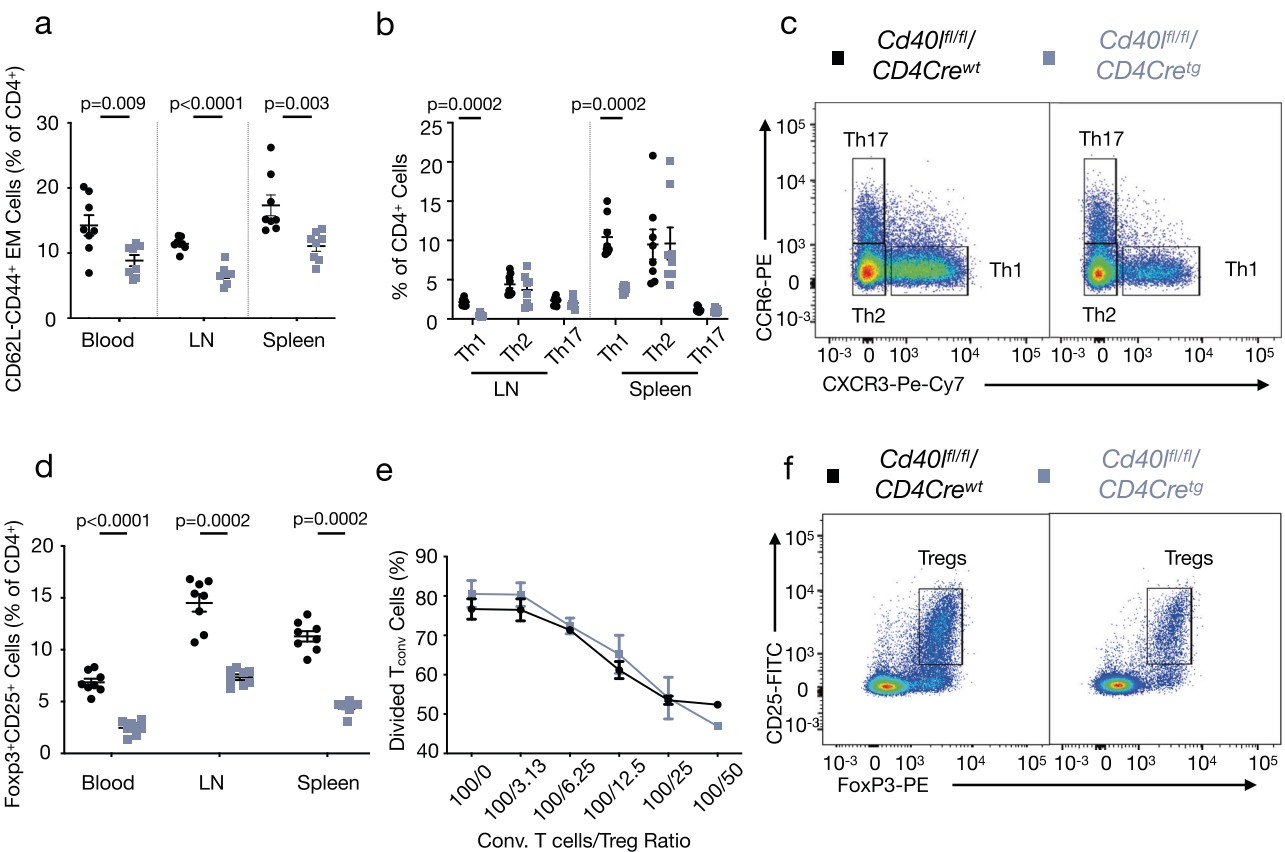

**Fig. 3 T cell CD40L deficiency leads to profound reductions in Th1 and Treg responses. a** Flow cytometric analysis of effector memory (EM) T cells in the blood [$n = 8$], LNs [$n = 8$], and spleen [$n = 8$]. **b** Flow cytometric analysis of effector T cell subpopulations including T helper (Th) 1 (CD62L$^-$CD44$^+$CXCR3$^+$CCR6$^-$), Th2 (CD62L$^-$CD44$^+$CXCR3$^-$CCR6$^-$), and Th17 (CD62L$^-$CD44$^+$CXCR3$^-$CCR6$^+$) in the LNs [$n = 8$] and spleen [$n = 8$]. **c** Representative gating strategy for effector T cell populations. **d** Flow cytometric analysis of Foxp3$^+$CD25$^+$ T regulatory cells (Tregs) in the blood [$n = 8$], lymph node (LN) [$n = 8$], and spleen [$n = 8$] in $Cd40l^{fl/fl}/Cd4Cre$ mice. Proliferation assay [$n$: WT = 3, TG = 4 for all data points except 100/50 condition where TG = 2] of Carboxyfluorescein succimidyl ester (CFSE)-stained CD4$^+$ T cells with varying ratios of conventional CD4$^+$ T cells to Tregs (ie. 100/0) analyzed by CFSE-dilution of conventional CD4$^+$ T cells. **f** Representative gating strategy for Tregs. All data are represented as mean ± s.e.m and $n$ refers to biologically independent animals. Data were analyzed by either a two-tailed unpaired Student's $t$ test (**a**, **e**) or two-tailed Mann–Whitney test (**b**, **d**). Source data are provided as a Source Data file.

atherosclerotic plaque phenotype with smaller necrotic cores and an increased fibrous cap thickness. In this T cell—DC axis, deficiency of CD40L-CD40 signaling hampers a Th1 response and particularly reduces IFN-γ levels. The CD40(L)-IFN-γ association was also present in human atherosclerosis, as both sCD40L and sCD40 levels in plaque and plasma showed a significant correlation with plaque and plasma levels of IFN-γ.

The pro-inflammatory and pro-atherogenic nature of Th1 cells, along with their prototypical cytokine IFN-γ, is well documented[2]. IFN-γ has been shown to activate and potentiate the expression of endothelial expressed adhesion molecules and pro-inflammatory chemokines allowing for the recruitment of leukocytes to the plaque[32]. In addition, IFN-γ impairs SMC-mediated collagen synthesis thus weakening of the fibrous cap[33]. Mice deficient in *IFN-γ* develop significantly smaller, more collagenous lesions resembling our phenotype[34]. Thus, our data strongly suggest that T cell CD40L primes atherosclerosis through IFN-γ activation, which initiates immune cell recruitment, necrotic core formation, and fibrous cap thinning.

Although T cell CD40L ablation induced a clinically favorable plaque phenotype, it also impaired Treg development without affecting Treg suppressive functions or apoptosis. The involvement of CD40L-CD40 interactions in Treg development has been reported before in global CD40$^{-/-}$ mice[35], and it was shown that CD40 expression on DCs, thymic epithelial cells, and thymic B

cells is critical for the in vivo generation and/or homeostasis of thymic Foxp3$^+$ Tregs[36,37]. However, how T cell CD40L deficiency contributes to Treg development exactly still needs to be elucidated. Tregs are powerful mediators of atherosclerosis[38], and clearly dampen the atherogenic immune response[39], and thereby prevent lesion growth. The first clinical trial treating CVD patients with low-dose IL-2 to specifically expand Tregs is currently ongoing[40,41]. Although Tregs are of major importance during atherogenesis, the reduction in Tregs that we observed in our T cell-deficient CD40L and DC-deficient CD40 mice did not result in aggravated atherosclerosis. On the contrary, blocking the T cell—DC CD40L-CD40 axis reduces atherosclerosis, revealing that a decreased Th1 response can overcome a reduction in immune-regulatory cells, most likely because less immune activation requires less immune suppression. While the reduced immune-suppressive requirement observed in our study may reduce the necessity for anti-inflammatory Tregs under atherosclerotic conditions, it is also important to note that T-cell subsets are highly plastic in terms of phenotypic characteristics during unresolving inflammation, especially in advanced stages of atherosclerosis which we have studied here. The number of Tregs was reported to be three-times higher in early than in advanced stages of atherosclerosis[42], indicating that Tregs exert most of their immune-regulatory effects in early atherosclerosis. Moreover, Tregs in more advanced stages of atherosclerosis exhibit

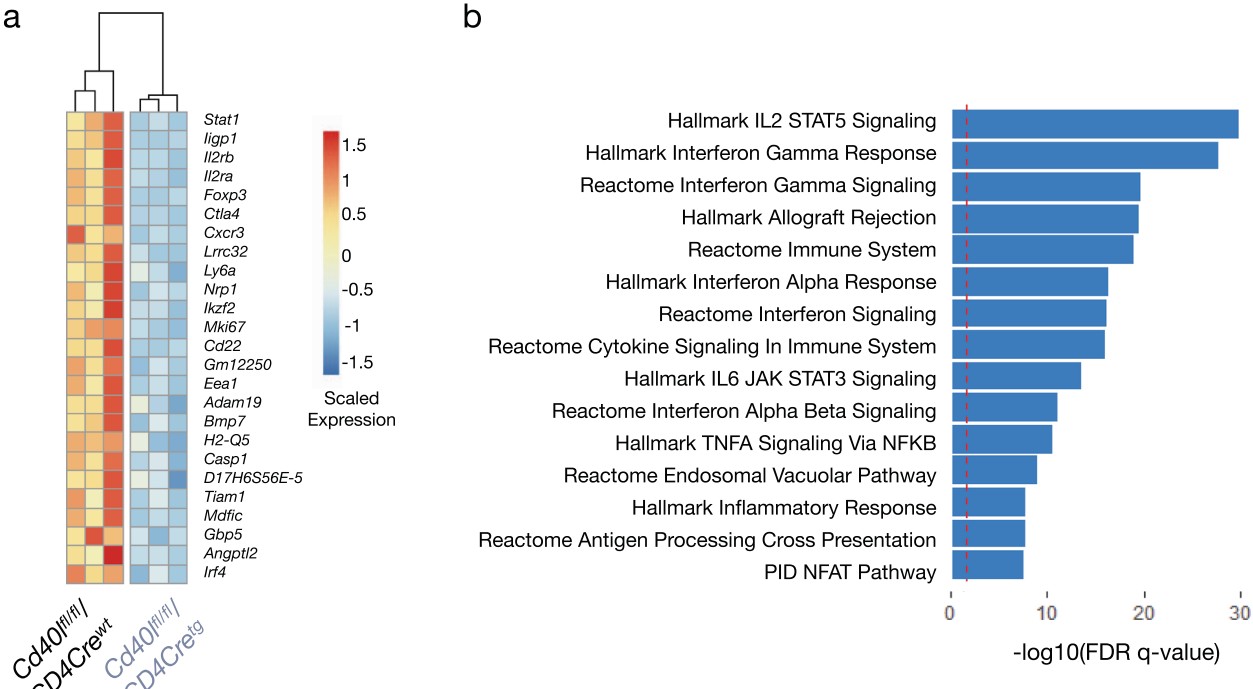

**Fig. 4 RNA sequencing confirms dysregulation of Th1 and Treg pathways in CD40L deficient T cells. a** Heatmap of top 25 downregulated genes in *Cd40l*[fl/fl]*/Cd4Cre*[tg] CD4+ T cells (right) compared to CD4+ T cells from their wild type littermates (left) [*n* = 3]. Data are normalized as reads per kilobase million with significance noted as a *p* adjusted value < 0.05 (corrected Benjamini–Hochberg). **b** Geneset enrichment analysis (GSEA) displaying the top 15 significantly enriched canonical and hallmark gene sets associated with downregulated genes in *Cd40l*[fl/fl]*/Cd4Cre*[tg] CD4+ T cells. *n* refers to biologically independent animals.

signs of plasticity. They co-express Foxp3 and IFN-γ and show a mixed Th1/Th17/Treg based transcriptional profile, undermining their protective nature[43]. As we did not see alterations in the suppressive capacities of CD40L-deficient Tregs decrease in vitro, we assume that the decrease in Th1 response in our T cell-specific CD40L (*Cd40l*[fl/fl]*/Cd4Cre*[tg]) deficient mice with advanced atherosclerosis affects atherogenesis to a larger extent than the decrease in, potentially partly dysfunctional, Tregs.

Another phenotypic aspect of our model that, besides, or even due to a decreased Th1 response, will have contributed to the observed decrease in atherosclerosis, is the reduction in IgG, and more specifically anti-oxLDL IgG antibodies. Germinal center derived IgG antibodies are known to promote atherogenesis and do this via multiple atherogenic pathways including metabolic reprogramming of immune cells, growth factors as well as immune cell recruitment[44]. In addition, antibodies against oxidation specific epitopes, including oxidized (ox)LDL also play a major role in atherosclerosis[45]. Whereas natural IgM oxLDL antibodies are considered to protect against atherosclerosis[46], via the induction of IL-5, IgG antibodies against oxLDL are considered pro-atherogenic[47]. The lack of Ig isotype switching, characteristic of CD40(L) deficiency, was also observed in our T cell deficient CD40L mice where total IgG, as well as oxLDL specific IgG, had decreased thereby contributing to the observed decrease in atherosclerosis.

Platelets are the other major CD40L expressing cell type, and they represent an important player in atherosclerosis. Not only are platelets considered to play a role in the development of atherosclerosis, i.e., in the recruitment of immune cells to the arterial wall and atherosclerotic lesion[48], they also play a pivotal role in atherothrombosis: intravascular damage caused by the rupture of an atherosclerotic plaque, which leads to massive occlusive platelet aggregation thereby causing tissue ischemia, clinically manifested as myocardial infarction or stroke[49]. Platelet

CD40L has been reported to be involved in several immunological aspects of platelet function. In platelets, CD40L can signal via its receptor CD40, and activate NFkB signaling via TRAF2, thereby contributing to platelet activation and release of inflammatory chemokines[50]. Platelet CD40L facilitates leukocyte recruitment to the arterial wall, the formation of platelet-leukocyte aggregates and platelet chemokine secretion[27], all mechanisms related to atherosclerosis development. However, the CD40-TRAF2 pathway only reflects a minor part of the CD40L cascade in platelets, and the interaction between CD40L and the integrin αIIbß3 appeared to be more relevant[26,51]. Deficiency of CD40L-αIIbß3 interactions resulted in less αIIbß3 activation, decreased thrombus growth, and increased thrombus stability, as also observed in the platelet-specific (*Cd40l*[fl/fl]*/Pf4Cre*) model in our study, which appeared independent of the receptor CD40[51]. In the present study, we did not observe any effect of selective deficiency of platelet CD40L on atherosclerotic plaque burden or phenotype. However, we did find a beneficial role for deficiency of platelet CD40L in an atherothrombosis model, confirming the importance of platelet CD40L in thrombus formation and stability. At first sight these findings seem contradictory to a previous study, where we demonstrated that repeated injection of thrombin activated CD40L-deficient platelets reduced atherosclerosis when compared to injection of thrombin activated wild type platelets by decreasing leukocyte recruitment, PLAs and thrombus formation[27]. However, these results actually show that platelet CD40L affects atherogenesis in models causing a certain level of platelet activation.

The data presented in this study supports and explains the differing roles of T cell and platelet CD40L in atherosclerosis. We show that the T cell-DC CD40L-CD40 plays a role in plaque progression and stability via the Th1-associated IFN-γ pathway while platelet CD40L regulates atherothrombosis. Taken together, these data stress the importance of unraveling cell type-specific

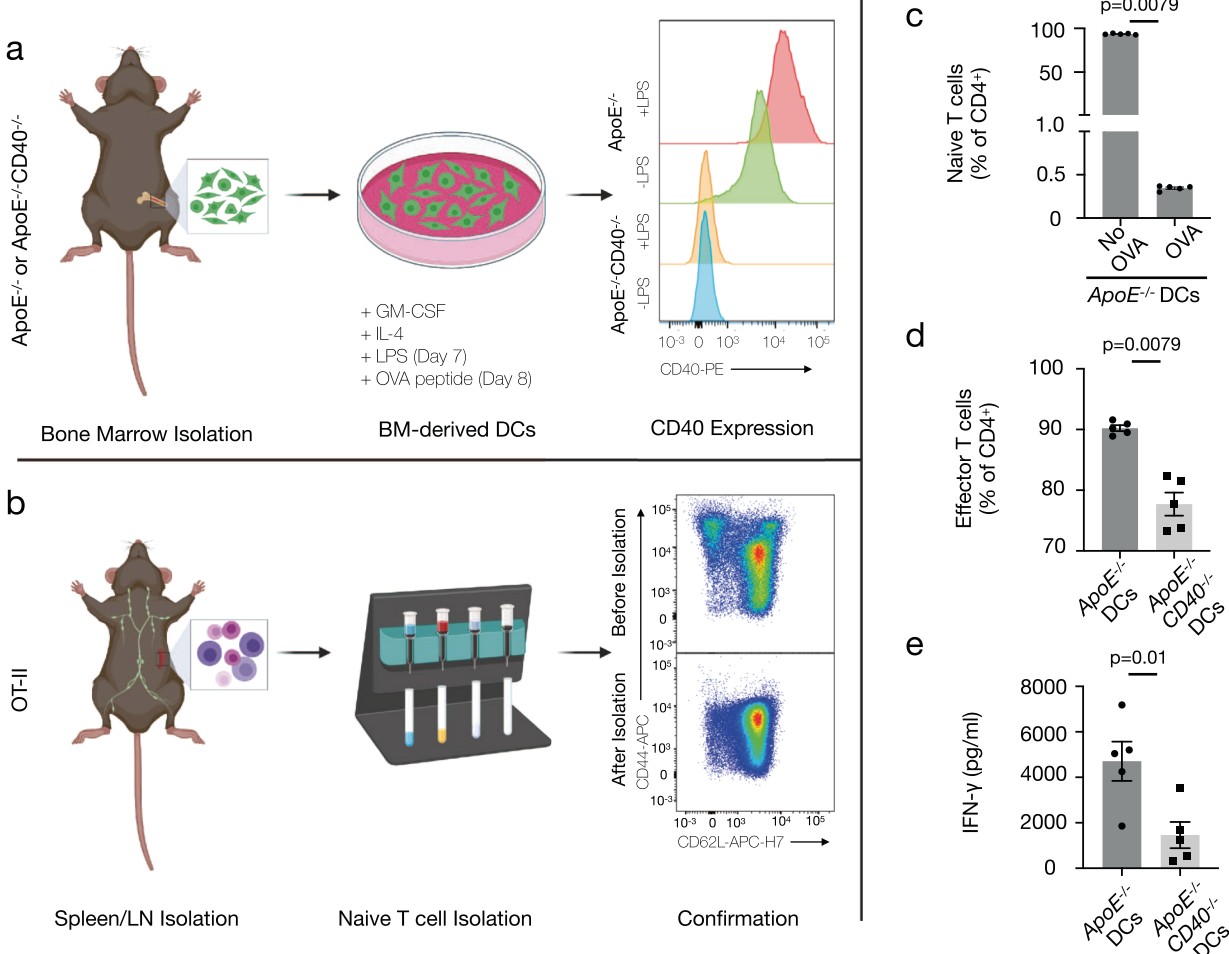

**Fig. 5 Disruption of the Dendritic Cell CD40-T cell CD40L axis results in lower interferon-γ production. a**, **b** Experimental setup for isolation and confirmation of (**a**) wild type and CD40-deficient dendritic cells (DCs) and (**b**) naive T cells for DC-T cell co-culture for in vitro disruption of the CD40-CD40L axis. Specifically, **a** bone marrow-derived DCs were generated from control *ApoE⁻/⁻* mice and *ApoE⁻/⁻* mice with a global deficiency of CD40 through polarization with Granulocyte-macrophage colony-stimulating factor (GM-CSF) and interleukin-4 (IL-4) and subsequent maturation with lipopolysaccharide (LPS). CD40 expression was measured by flow cytometry. **b** Naive T cells (CD62L⁺CD44⁻) were isolated from the lymph nodes (LN) and spleens of OT-II transgenic mice with a T cell receptor specific for ovalbumin (OVA) peptide 323–339. Mature DCs were stimulated with OVA peptide 323–339 and cultured with naive OT-II T cells for 72 h for antigen presentation and T cell stimulation. **c** Mature *ApoE⁻/⁻* DCs were stimulated with the OVA peptide or a vehicle control [n = 5] to confirm the specificity of the T cell receptor on the OT-II T cells. **d** Flow cytometric analysis of effector T cell (CD62L⁻CD44⁺) populations following DC-T cell co-culture [n = 5]. **e** Supernatant concentration of interferon-γ (IFN-γ) following DC-T cell co-culture [n = 5]. Data are represented as mean ± s.e.m and n refers to biologically independent animals. Data were analyzed by either a two-tailed unpaired Student's t test (**e**) or two-tailed Mann–Whitney test (**c**, **d**). Source data are provided as a Source Data file.

actions of promising immunotherapeutic targets in disease. Although antibody mediated global inhibition of CD40L seemed a promising therapy, this resulted in an increased risk for thrombo-embolic complications, as the effects of platelet CD40L inhibition were not taken into consideration[17]. On the other hand, targeting CD40-TRAF6 signaling, which is the predominant CD40 signaling pathway in macrophages, did decrease atherosclerosis without causing immunosuppressive side effects, and is an attractive strategy to combat atherosclerotic CVD[52–54]. By deciphering the cell type-specific effects of the CD40L-CD40 pathway in disease, targeted therapeutic strategies directed against cell types (i.e., via cell-specific bi-specific antibodies[55] or targeted nanobiologics[56]) or cell type-associated signaling intermediates, we will be able to more precisely target the desired effector pathways of the CD40L-CD40 dyad, thereby limiting unwanted side effects.

## Methods

**Mice**. For the atherosclerosis studies, *Cd40l^fl/fl* mice were successfully generated via insertion of loxP sites flanking exon 3, which enabled cre-mediated deletion of exon 3 causing a translational frameshift rendering downstream exons nonfunctional (Ozgene Pty Ltd, Bentley, Australia, Supplementary Fig. 1a)[57]. *Cd40^fl/fl* mice were successfully generated via insertion of loxP sites flanking exons 2 and 3, which enabled cre-mediated deletion of exon 2 and 3 (Ozgene Pty Ltd, Bentley, Australia, Supplementary Fig. 2a)[58]. Cell-specific knockouts were generated by backcrossing *Cd40^fl/fl* and *Cd40l^fl/fl* mice to either *CD11cCre* (stock No: 008068, Jackson Laboratory, Bar Harbor, Maine, USA)[59], *Cd4Cre* (stock No: 017336, Jackson Laboratory, Bar Harbor, Maine, USA)[60] or *Pf4Cre* (stock No: 008535, Jackson Laboratory, Bar Harbor, Maine, USA)[61] transgenic mice. Finally, all strains were backcrossed at least ten times to *Apoe⁻/⁻* transgenic mice (stock No. 002052, Jackson Laboratory, Bar Harbor, ME, USA) to generate atherosclerotic DC-specific CD40 deficient mice or T cell and platelet specific CD40L deficient transgenic mice.

Male *Cd40l^fl/fl*/*Cd4Cre^tg* (n = 17–19), *Cd40l^fl/fl*/*Pf4Cre^tg* (n = 19–21) and *Cd40^fl/fl*/*Cd11cCre^tg* (n = 10–15) along with their respective Cre^wt littermates (all on an *ApoE⁻/⁻* background) were administered a normal chow diet for 28 weeks. At 28 weeks of age, mice were euthanized following an intraperitoneal injection with Ketamine/Xylazine.

For the in vitro DC-T cell co-culture, mice with a global deficiency of CD40 ($CD40^{-/-}$) were generated by insertion of *neo* resistance cassette into exon 3, which interrupted the coding sequence[31]. $CD40^{-/-}$ mice were backcrossed to an $ApoE^{-/-}$ background for at least ten generations to generate $ApoE^{-/-}CD40^{-/-}$ mice. Male $ApoE^{-/-}$ mice (stock No. 002052, Jackson Laboratory, Bar Harbor, ME, USA) were used as a control for CD40 wild type mice. T cells were isolated from two month old male OT-II transgenic mice (stock No. 004194, Jackson Laboratory, Bar Harbor, ME, USA) with T cell receptor specific for the OVA residue 323–339 peptide ($n = 5$)[62]. Mice were bred and housed at the animal facility at Ludwig-Maximilians University of Munich following institutional guidelines. All animal experiments were approved by the local ethical committee for animal experimentation (TVA #55.2-1-54-2532-85-2014).

**Hematology and organ isolation**. Blood was obtained via cardiac puncture and collected into EDTA-containing tubes (Sarstedt, Nümbrecht, Germany). Hematological parameters were determined using a ScilVetabc plus (Scil Animal Care Company B.V., Viernheim, Germany). Following perfusion of the arterial tree with phosphate-buffered saline (Sigma Aldrich, St. Louis, USA) and nitroferricyanide (III) dehydrate (Sigma Aldrich), the abdominal aorta, aortic arch, liver, lymph nodes, and spleen were harvested. For gene expression analysis, organs were stored in RNAlater (Life Technologies, Carlsbad, USA) for 24 h at room temperature and subsequently frozen at −80 °C. For flow cytometric analysis, organs were collected in phosphate-buffered saline on ice. To analyze plaque development, hearts including the aortic root were isolated and frozen in Tissue-Tek (Sakura, Finetek, Torrance, USA) for sectioning.

**Morphometry and histology**. Beginning from the onset of the aortic valves, hearts were cut in 8 µm-thick sections until the valves disappeared. Serial sections were stained with Oil-red O (Sigma Aldrich) and images were recorded using a brightfield microscope. Analysis was performed using an automated morphometry system (LAS 4.6 analysis, Leica Microsystems) on a Leica DM6000 microscope (Leica Microsystems, Wetzlar, Germany).

**Immunohistochemistry**. Frozen cryosections from the aortic root were fixed in ice-cold acetone prior to incubation with primary antibodies (see Supplementary Table 3 for details). Fluorochrome-conjugated secondary antibodies (Alexa Fluor 488, Alexa-594, Cy3, Life Technologies and Jackson ImmunoResearch, West Grove, USA) were used to detect binding of primary antibodies. Tissues sections were counterstained with hematoxylin or 4′,6′Diamidino-2-phenylindol (DAPI, Life Technologies), mounted with DAKO fluorescent mounting medium (Dako, Agilent Technologies, Santa Clara, USA), and recorded using a Leica DM6000 microscope. CD3, CD4, and Foxp3 stained cells were counted while Mac3 and α-SMA positive areas were analyzed by applying color thresholds.

**Plasma cholesterol, antibody titers, and sCD40L**. Plasma was isolated by centrifugation (500 × *g*, 15 min, 4 °C) of EDTA-anticoagulation whole blood. To measure cholesterol, plasma was diluted 1:5 and analyzed using a colorimetric assay (CHOD-PAP, Roche Diagnostics, Basel, Switzerland). Total Ig levels were measured using a multiplex bead-based assay (Mouse Ig Isotyping Panel 6plex, EBiosciences) while anti-oxLDL titers were measured using an enzyme-linked immunosorbent assay (ELISA). All ELISA antibodies were obtained from Jackson ImmunoResearch Labs and handled according to manufacturer's data sheets. Soluble CD40L (sCD40L) was measured in the plasma using an sCD40L sandwich ELISA according to the manufacturer's instructions (ab119517, Abcam).

**Flow cytometry**. Harvested spleens and lymph nodes were torn apart manually using forceps and subsequently filtered through a 70 µm stranger in order to prepare single cell suspensions. To lyse red blood cells, suspensions containing whole blood and spleens were first incubated with a red blood cell lysis buffer containing 150 mM ammonium chloride (Sigma Aldrich) and 10 mM sodium bicarbonate (Sigma Aldrich) at pH 7.4 for 2 min on ice. After incubation, cells were washed with PBS and plated for staining in microtiter plates (Costar 3799, Corning, Corning, USA). When necessary, cells were first stained with Fc-block (anti-CD16/32, eBiosciences, clone 93, 1:100) for 20 min on ice to prevent nonspecific binding. After washing, cells were stained with different combinations of antibodies depending on the experiment (see Supplementary Table 4 for clones). Viability dyes (Live/Dead fixable Aqua/Violet/Near-Infrared; Life Technologies) were included in antibody panels when necessary. For example, apoptosis was determined by positive staining for fluorochrome-conjugated annexin A5 (Biolegend) with simultaneous exclusion of dead cells determined by Live/Dead staining. Single cell suspensions were analyzed using a BD FACS Canto™ II (BD Biosciences) and data were analyzed using Flowjo v.10 software (Flowjo, LLC, Ashland, USA).

**Dendritic cell-T cell co-culture**. After euthanasia, bone marrow was collected from the femurs of male $ApoE^{-/-}$ and $ApoE^{-/-}CD40^{-/-}$. Briefly, bones were cleaned of any muscle and other tissue then collected in serum-free RPMI 1640 (Gibco, Thermo Fisher Scientific). The femur was cut near the knee joint and centrifuged at $9300 \times g$ for 1 min at 4 °C. The bone marrow containing pellet was

washed with PBS and centrifuged at $300 \times g$ for 5 min at 4 °C. Cells were incubated with a red blood cell lysis buffer containing 150 mM ammonium chloride (Sigma Aldrich) and 10 mM sodium bicarbonate (Sigma Aldrich) at pH 7.4 for 1 min on ice. After incubation, cells were washed with RPMI1640 plus Glutamax (Life Technologies) media supplemented with 10% fetal bovine serum (FBS, Life Technologies), 100 U/ml Pencillin and 100 mg/ml Streptomycin (P/S, Life Technologies), and 50 uM ß-mercapatoethanol (Sigma Aldrich), and 20 ng/ml recombinant GM-CSF (Peprotech) termed R10 media. 20 million per animal were plated in 100 mm non-tissue culture treated dishes in 20 ml of R10 media. After 3 days, 20 ml of R10 media was added. On Day 6, 20 ml of culture media was removed centrifuged at $300 \times g$ for 5 min at 4 °C to recover non-adherent cells. After centrifugation, the supernatant was aspirated, and cells were resuspended in fresh R10 media supplemented with 10 ng/mL recombinant IL-4 (Biolegend). On Day 7, the cell culture supernatant was aspirated to obtain the non-adherent DCs. Adherent cells including the bone marrow derived macrophages were not used for co-culture. Cells were centrifuged as described earlier and resuspended at a concentration of 1 million cells/ml in R10 media supplemented with 5 ng/ml IL-4 and 1 ug/ml LPS to upregulate MHCII and CD40 expression. After overnight stimulation, cells were washed with PBS to remove residual cytokines. Dendritic cells (DC) were resuspended in R10 media with 10 ug/ml 323–339 OVA peptide (Sigma Aldrich) for 1 h at 37 °C.

To isolate naïve T cells, the spleen and axillary, brachial, and inguinal lymph nodes were harvested from OT-II mice after euthanasia. Harvested spleens and lymph nodes were torn apart manually using forceps and subsequently filtered through a 70 µm stranger in order to prepare single cell suspensions. To lyse the red blood cells, single cell splenic suspensions were first incubated with a red blood cell lysis buffer containing 150 mM ammonium chloride (Sigma Aldrich) and 10 mM sodium bicarbonate (Sigma Aldrich) at pH 7.4 for 2 min on ice. After incubation, cells were washed with PBS and counted (TC-20, Biorad). 50 million cells were aliquoted and centrifuged at 400 x *g* for 5 min at 4 °C. Naïve T cells were isolated according to the manufacturer's instructions using the mouse naïve CD4 + T cell isolation kit (Miltenyi Biotec). After isolation, T cells were resuspended with RPMI1640 plus Glutamax (Life Technologies) media supplemented with 10% fetal bovine serum (FBS, Life Technologies), 100 U/ml Pencillin and 100 mg/ml Streptomycin (P/S, Life Technologies), 1× MEM non-essential amino acids (Sigma Aldrich), 1 mM sodium pyruvate (Sigma Aldrich) and 50 µM 2-mercaptoethanol (Sigma Aldrich). 100,000 naïve T cells were aliquoted in duplicate in a round bottom, tissue culture treated 96 well plate with 20,000 stimulated DCs (1:5 ratio). Dendritic cells (DC) and T cells were cultured together for 72 h before cells were prepared for flow cytometry as described in the flow cytometry section. Culture supernatant was frozen at −80 °C until it was used following manufacturer's instruction using a multiplex bead-based assay (Mouse Th1/Th2/Th2/Th17/Th22/Treg 17-plex panel, Thermo Fisher Scientific).

**Proliferation and suppression assays**. Splenic CD4+ T cells were negatively selected using antibody-conjugated magnetic beads according to manufacturer's instructions (Dynabeads Untouched Mouse CD4, Life Technologies). Isolated CD4+ T cells were stained in a 3 µM carbofluorescein succinimidyl ester (CFSE, Life Technologies) solution for 10 min to follow cell proliferation. After staining, cells were cultured in RPMI1640 plus Glutamax (Life Technologies) media supplemented with 10% fetal bovine serum (FBS, Life Technologies), 100 U/ml Pencillin and 100 mg/ml Streptomycin (P/S, Life Technologies), 1× MEM non-essential amino acids (Sigma Aldrich), 1 mM sodium pyruvate (Sigma Aldrich) and 50 µM 2-mercaptoethanol (Sigma Aldrich). Anti-CD3/CD28 antibody-conjugated magnetic beads were added to the culture according to manufacturer's instruction to induce proliferation. The protocol for the Treg suppression assay paralleled the T cell proliferation with the addition of varying concentrations of Tregs added to the culture. To isolate Tregs, $CD3^+CD4^+CD25^{hi}$ cells were sorted via flow cytometry (BD FACS Aria III, BD Biosciences). Cells were allowed to proliferate for 72 h in culture. Differences in proliferation were determined by CFSE-dilution as measured by flow cytometry (BD FACS Canto II, BD Biosciences). Data were analyzed using Flowjo v.10 (Flowjo, LLC, Ashland, USA).

**Real-time PCR**. Organs stored in RNAlater (Life Technologies) at −80 °C were thawed and lysed in TRIzol (Life Technologies) using 7 mm steel beads in a TissueLyser (Qiagen). RNA was isolated using the RNeasy Mini Kit II (Qiagen) and reverse transcribed with the SuperVilo cDNA synthesis kit (Life Technologies). Quantitative PCR (qPCR) was performed using TaqMan gene expression assays (see Supplementary Table 5 for assay details) on a 7900T Fast Real Time PCR system (Applied Biosystems, Thermo Scientific). Expression was normalized using *18 s rRNA* as a housekeeping gene. Relative quantification of gene expression was analyzed using the $2^{-\Delta\Delta CT}$ method.

**RNA-sequencing**. Splenic CD4+ T cells were negatively selected using antibody-conjugated magnetic beads according to manufacturer's instructions (Dynabeads Untouched Mouse CD4, Life Technologies). Total RNA was isolated from the CD4+ T cells using the RNeasy Mini Kit II (Qiagen) with an on-column DNase I treatment. RNA concentration was determined using a Qubit fluorometer (Qiagen) while RNA quality was evaluated using an Agilent Biolanalyzer 2100 system (Agilent

Technologies, CA, USA) prior to library preparation. 100 ng of total RNA was used to construct strand-specific libraries using the Ovation RNA-Seq System (NuGen Technologies). Libraries were pooled and diluted to 10 nM prior to sequencing on a HiSeq 4000 (Illumina) at a depth of 20 million single ended 50 base pair reads. Reads were aligned to the mouse genome mm10 using STAR 2.5.2b with default settings. BAM files were indexed and filtered on MAPQ > 15 with SAMtools 1.3.1[63]. Raw tag counts and RPKM (reads per kilobase per million mapped reads) values per gene were summed using HOMER2's analyzeRepeats.pl script with default settings and the -noadj or -rpkm options for raw counts and RPKM reporting, respectively[64]. Differential expression was assessed using the DESeq2 (1.20.0) Bioconductor package in an R 3.5.2 environment with a Benjamini–Hochberg corrected $p$ value < 0.05 and an average RPKM > 1 in at least one group[65]. To identify deregulated pathways, gene ontology (biological process) and pathway enrichment was performed using the EGSEA (1.6.1) Bioconductor package as well as Ingenuity Pathway Analysis (Qiagen)[66].

**Micro-spot platelet aggregation.** Blood was collected from retro-orbital puncture into 40 μM D-phenylalanyl-prolyl-arginyl chloromethyl ketone, 5 U/ml heparin, and 40 U/ml fragmin. Samples of 400 μl were preincubated with $DiOC_6$ (0.5 μg/ml, Anaspec, Fremont, CA, USA) and fibrinogen AF546 (25 μg/ml, Thermo-Fischer Scientific, Waltham, MA, USA) for 5 min before whole blood perfusion. Coverslips coated with micro-spots of type I collagen (1 μl, 100 μg/ml, Nycomed, Hoofddorp, The Netherlands) were mounted on a transparent, parallel plate flow chamber (50 μm depth, 3 mm width, and 300 mm length). Flow perfusion using a shear rate of $1000\ s^{-1}$ was performed on all samples[67]. Brightfield and fluorescence images were captured every 30 s for 3.5 min using an EVOS microscope, equipped with a 60× objective. Surface area coverage was analyzed using semi-automated scripts operating in Fiji (ImageJ)[68].

**Atherothrombosis.** The endothelium of mouse carotid arteries was completely denuded using an angioplasty catheter guide. Briefly, 8- to 12-week-old $Cd40l^{fl/fl}/Pf4Cre$ mice (body weight 18–23 g) were fed a western diet for seven days before and 28 days after wire injury of the common carotid artery. After intraperitoneal anesthesia (medetomidine (0.5 mg/kg), midazolam (5 mg/kg) and fentanyl (0.05 mg/kg)) midline neck incision was used to expose the left carotid artery and branches. A 7/0 surgical suture was looped around the distal left common carotid artery and proximal internal and external carotid arteries to temporary interrupt perfusion. An additional suture was looped around the distal external carotid artery near the bifurcation followed by a small hole insertion on the external artery between the sutures before inserting a flexible angioplasty guide wire (0.36 mm) into the common carotid artery (7–8 mm). Endothelial denudation was performed by three passes along the vessel with rotation. The wire was removed, and the proximal and distal sutures were tied off to restore blood flow. The skin incision was closed, and anesthesia was antagonized by intraperitoneal application of atipamezol (2.5 mg/kg), flumazenil (0.5 mg/kg), and buprenorphine (0.1 mg/kg) followed by analgesic therapy with subcutaneous meloxicam (0.2 mg/kg) 4–6 h after surgery as well as two more injections in an interval of 24 h.

**Assessment of CD40, CD40L and IFN-γ in human plasma and in atherosclerotic plaques.** Blood samples and plaque tissue were obtained from 185 subjects of the Carotid Plaque Imaging Project (CPIP, Lund University) cohort, which consists of patients undergoing carotid endarterectomy at the Vascular Department of Skåne University Hospital (Malmö, Sweden). Samples were collected between 2005 and 2010. Written informed consent was given by all patients and the study protocol was approved by the local Regional Ethical Committee (reference number 472/2005). The study fully conforms to the Declaration of Helsinki.

Indications to surgery were as described previously:[69] patients with ipsilateral symptoms (amaurosis fugax, transient ischemia attack or stroke within 6 months prior to surgery) and a degree of stenosis >70% or without ipsilateral symptoms but with a degree of stenosis >80%. The degree of stenosis was assessed with doppler ultrasound based on flow velocities as previously validated[70]. All patients were evaluated by a neurologist prior to acceptance for surgery. Clinical characteristics of the patient population are summarized in Supplementary Table 2. Blood samples were taken the day before endarterectomy. Immediately upon surgical removal plaques were snap-frozen in liquid nitrogen and plaque homogenates were prepared as previously described[71]. Briefly, 2 mM thick fragments were removed for histology. Plaques were weighed before being homogenized at 1600 rpm with a motorized blender in 5 mL of buffer containing 50 mmol/L Tric-HCl, 0.25 mol/L sucrose, 2 mmol/L tris (2-carboxyethyl)phosphine HCl. 50 mmol/L NaF, 1 mmol Na-orthovanadate, 10 mmol/L Na-glycerophosphate, 5 mmol/L Na-pyrophosphate, protease inhibitor cocktail, 1 mmol/L benzamidine, and 10 mmol/L phenylmethylsulfonyl fluoride.

CD40 and CD40L were analyzed in plaque tissue homogenates and in plasma samples using the Proximity Extension Assay (PEA) technique using the Proseek Multiplex CVD96x96 reagents kit (Olink Bioscience, Uppsala, Sweden)[72]. IFN-γ levels were determined in plaque tissue homogenates and in plasma samples using multiplex analysis (Luminex).

**Statistical analysis.** All data are expressed as mean ± standard error of the mean (s.e.m.) and statistical analysis was performed with Graphpad Prism v.7 (Graphphad Software Inc., La Jolla, USA). After testing for normality using Anderson-Darling (A2) normality test, data were analyzed by two-tailed unpaired Student's $t$ test or Mann–Whitney test. For analysis of Virmani classification of plaques, data were analyzed with Pearson's chi-squared test. Differences with $p$ value of <0.05 were considered statistically significant.

Correlations between CD40(L) and IFN-γ measurements in human plasma and plaque tissue were analyzed using Spearman's rank correlation. All continuous variables describing clinical patient characteristics were found to be skewed (using Kolmogorov–Smirnov and Shipiro–Wilk tests to assess Gaussian distribution) and are shown as median with interquartile range.

**Reporting summary.** Further information on research design is available in the Nature Research Reporting Summary linked to this article.

## Data availability

The data supporting the findings of this study are available within the paper and its Supplementary Information files. Due to regulations of the Swedish law, patient related data, even when unidentified by code, is considered sensitive data as long as the subjects are alive and is not allowed to be shared as individual data points. RNA sequencing data are publicly available under accession code [GSE145585]. Source data are provided with this paper. All other data are available from the corresponding authors upon request.

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

## Acknowledgements

This study was supported by the Deutsche Forschungsgemeinschaft [CRC 1123 to D.A., E.L., N.G., S.S., R.M., C.W., TRR259 to N.G & E.L]. We also acknowledge the support from the Netherlands CardioVascular Research Initiative: the Dutch Heart Foundation, Dutch Federation of University Medical Centres, the Netherlands Organization for Health Research and Development and the Royal Netherlands Academy of Sciences for the GENIUS-II project "Generating the best evidence-based pharmaceutical targets for atherosclerosis" [CVON2017-20]. This study was also supported by the Netherlands Organization for Scientific Research (NWO) [VICI grant to E.L.]; the EU (Horizon 2020, REPROGRAM to E.L.); the German Centre for Cardiovascular Research (DZHK) [High-risk high-volume (HRHV) grant to E.L, D.A. and C.W.] and the European Research Council [ERC consolidator grant to E.L, ERC advanced grant to C.W.]. C.W. is a Van de Laar professor of atherosclerosis. Further support was received from the Swedish Research Council (A.E, I.G), Swedish Heart and Lung Foundation (A.S, A.E, I.G), Swedish Society for Medical Research (A.E), Swedish Heart and Lung Association (A.S), Swedish Stroke Association (A.S.) and the Swedish Foundation for Strategic Research Dnr IRC15-0067. The Knut and Alice Wallenberg foundation, the Medical Faculty at Lund University and Region Skåne are acknowledged for generous financial support. Graphics for Fig. 5a and b were created with BioRender.com.

## Author contributions

E.L., N.G., D.A., and C.W. conceived the study. S.S., N.G., D.A., and E.L. contributed to the design and implementation of the research. M.L., C.B., H.W., K.N., T.T.P.S., M.E.R., Y.W., S.U., N.G., and D.A. performed experiments on DC and T cell specific knockout models. M.L., C.B., M.A., E.K., S.L.N.B., M.J.E.K., and J.W.M.H. performed experiments on the platelet specific knockout model. C.M.v.T. and K.H.M.P. prepared libraries for RNA sequencing and processed the data. R.M. and P.J.H.K. performed imaging experiments. A.S., A.E., and I.G. contributed to the design, implementation, and analysis of the human cohort. M.L., D.A., and E.L. wrote the paper.

## Competing interests

The authors declare no competing interests.
