## [Peer Review File · Nature Communications]

Reviewers' comments:

Reviewer #1 (Remarks to the Author):

Dr. Lutgens and colleagues have published numerous reports on the role of CD40-CD154 interactions in a mouse model of atherosclerosis. In this latest report, they focus on source requirements for CD154 in driving disease in this model. Overall, the work presents new data addressing the long-standing controversy regarding the physiologic role of platelet vs. T cell-expressed CD154 in stimulating CD40 signaling in various scenarios. However, some aspects of the work are not well-presented and/or discussed, and there is a tendency throughout to over-interpret results.

In several regards, the Introduction is confusing and not entirely accurate. In the final paragraph of p. 3, Ref. 17 is focused upon TRAF6, which is downstream of CD40 signaling, not a 'signaling intermediate' of CD154. It's important to be specific about when the authors are discussing CD154 as a ligand for the signaling receptor CD40, rather than signaling to the cell via CD154 itself (which has been suggested, although it's controversial). Here, the authors seem to be suggesting that CD154 itself is an important signal transducer in atherosclerosis, but the signals in this study all seem to be sent through CD40, not CD154. And CD40 'signaling intermediates' are indeed well-characterized, in contradiction to the statements in lines 90-92, which are confusing. The statement in lines 97-98 is also misleading. It is well-established that CD40 signaling cooperates with BCR signaling, but CD40 does not itself 'stimulate' either BCR or TCR signaling. In line 100, the phrase "pro-atherogenic subsets, including....." implies that >1 T cell subset has been implicated in both pro and anti-atherogenic activity, but only one subset is listed for each. If there are others, list them; if not, remove the plural form and the 'Including'. The sentence in lines 104-106 was very awkward and confusing. Please re-word this. Finally, the Introduction barely mentions the long-standing controversy regarding the true physiologic role(s) of platelet CD154, which is a key topic addressed in this study. This should be discussed in the Introduction.

1. The mouse models used here have not all been referenced or adequately described. What was the source of CD11cCre-Apoe^{-/-}, Cd4Cre-Apoe^{-/-} and Pf4Cre-Apoe^{-/-} mice that were backcrossed to the CD40^{fl/fl} and CD154^{fl/fl} mice that the authors had produced for them? Either cite references in which those mice were characterized, or present data on their validation here. Importantly, none of the data presented show the level of normal CD154 expression on mouse platelets, nor the effectiveness of removal of this expression – only data on T cell and DC-specific knockout mice are presented. It is critical to also validate the platelet-specific knockout mouse, to accurately interpret results in this report.

2. Although presented as statistically significant, some of the differences shown in the Figures are not biologically convincing, and are over-interpreted. For example, the differences in Fig. 3d are barely detectable, but described as a 'profound reduction' in EM T cells on p. 7. In this Figure, actual cell numbers – not just percentages – should also be shown. Lines 182-184 of this page describe a 77% reduction in Th1-produced IFN-g, but I could not find these data anywhere in the Figures. Either present the data, or remove their discussion from the text. Presumably the important site of action of T cells in this model is in the vasculature, not the spleen or LN – this needs to be considered when interpreting these data.

3. Interpretation of data presented in Fig. 5 is confusing. How do the authors reconcile biologically decreased inflammation in 2 models in which there are fewer Treg? I don't follow the authors' conclusions about OX40L expression; the only data addressing this are in 5f, and I see no difference – listing a p value is not meaningful when the difference is undetectable. A statistical difference is not necessarily a biologically important difference. For both these data, and those mentioned above in Fig. 3, the discussion needs to acknowledge the magnitude of the differences seen, which is quite modest.

4. Regarding data presented in Fig. 6, multiple published reports many years ago established a major defect in Ig isotype switching in the absence of CD154 (in both mice and people), so these results are hardly novel or surprising. The results do not establish that alterations in Ig production 'contributes' to the anti-inflammatory phenotype in the atherosclerosis model; this is a correlation,

for which cause and effect is not tested here.

5. The same tendency to equate correlation with causality is found in lines 255-256 of the Discussion. The authors cannot claim that inhibiting the Th1 response driven by CD154 is the major mechanism by which atherosclerotic plaques are stabilized, when this causal relationship has not been tested; right now, it is a correlation.

6. What do the authors suggest is the source of the striking but anomalous Treg numbers, as mentioned above, which seem counter-intuitive with the anti-inflammatory effect? This should be addressed in the Discussion. The Discussion should also compare and contrast results obtained with the conditional knockout mice here with those obtained earlier by this group in the whole-mouse CD40-deficient knockout (e.g. Ref. 10). Finally, the suggestion that the results presented here would allow cell-type-specific blocking of the CD40-CD154 interaction for therapy in vivo is baffling – how, exactly, do the authors envision this would be accomplished?

Minor comment:

The authors use past vs. present tense seemingly randomly when describing previously-published work. I believe the convention is to use present tense for published data; past tense for unpublished results (such as the current report). Please be consistent throughout the manuscript.

Reviewer #2 (Remarks to the Author):

This is an interesting and ambitious paper examining the role of CD40L-CD40 axis in atherosclerosis. Disruption of CD40 axis in T cells and in dendritic cells likely decreased atherosclerosis while disruption of platelet axis had no effect. These are interesting observations but lack of crucial experimental details does not allow confidence in the reported results at this stage. I would want to see convincing evidence of validation of the experimental models before accepting these results—in particular because some of the conclusions are at odds with existing paradigms—for example: the authors report that diminished Treg numbers are associated with decreased lesions –which the authors themselves recognize is counterintuitive. (Of course, other aspects are changed and these observations may be valid in context—but need to be sure models are appropriate).

Major:

The two most important methodological concerns are the genetic background of the mice and the adequacy of the various gene knockdowns in these models

First: It is well known in the atherosclerosis field that the genetic background of the mice can impact atherogenesis and that all experimental models must be on similar homogenous genetic backgrounds—which ideally should be at least 10 generations, but at minimum at least 7. There is no description of the genetic background of the floxed mice, the Cre mice and no indication of how many generations were these mice crossed to the apoE mice. —eg for the CD40Lfl/fl and the CD40fl/fl—what were their genetic backgrounds vs the apoE—how many generations were these backcrossed to the apoE? Similarly-what were the genetic backgrounds of the CD11cCre and the CD4Cre and Pf4Cre? These were said to be on apoE background but how many generations were each?

Second: The extent of the various knockdowns is not at all clear? For example—how does Suppl Fig 1b and c show knockdown of CD40L in T cells –fig b is a functional assay of cells in artery and the fig c seems to show # (or mfi) of CD4 or CD8 T cells vs B—how does this show deletion or knockdown of CD40L expression in respective T cells—these cell types should be isolated and extent of knockdown (eg residual gene expression) documented directly to demonstrate efficiency of knockdown to be convincing.—The CD40 kd in Suppl fig 2 should be studied in similar manner—the data shown only speaks to relative cell # as I understand. Finally—I don't see any data supporting the validation of the platelet specific deletion of Cd40L.

Minor:

Page 6—IL-6 and INF transcripts were measured in aorta—what does this mean if athero was decreased—less cells—what was denominator? What do the statistics refer to. You likely measured “several” transcripts- what other cytokines did you measure. Were others not changed—this information would be informative

The reduction in total IgG2b is interesting, consistent with reduced T help, but why do you think IgE and IgM decreased? The role of IgG to OxLDL in atherogenesis is controversial but IgM to OxLDL is most likely protective and due to innate and not adaptive (at least directly) mechanisms, perhaps consistent that despite disruption of T cell help, IgM to OxLDL did not change.

Table 1: The chol levels shown in Table 1 appear to be wrong units 3-4 mg/dL (? mmol/L) These also very low levels for apoE mice and too low to cause extensive atherogenesis
The chol levels of each group at the end of the 28 diet period should be shown—and ideally at the beginning. As these are apoE mice—the TG levels should be shown

General comment: While these observations may to help explain the pathophysiology of atherogenesis and the possible complex roles of CD40L-CD40 interactions, at a practical level they do not likely offer specific targets for therapeutic interventions in the context of altering atherogenesis due to their impact on other immunological processes and autoimmunity. My own suggestion would be to refocus the Discussion on how knowledge of these interactions in atherosclerosis might help inform the impact of interfering on this pathway in other diseases.

Reviewer #3 (Remarks to the Author):

In this study, Lacy et al. address the role of the CD40-CD40L axis in atherosclerosis. To this aim, the authors have generated T cell- and platelet-specific CD40L-deficient and DC-specific CD40-deficient mice, and backcrossed these conditional mutants onto an atherosclerosis prone ApoE^{-/-} background. At 28 weeks of age, both T cell-specific CD40L and DC-specific CD40 deficiency reduced plaque progression, associated with hampered Th1 polarization, impaired antigen-dependent T cell proliferation and reduced oxLDL IgG production. In contrast, platelet-specific CD40L deficiency failed to decrease atherosclerosis, but ameliorated atherothrombosis. Altogether, this work represents a largely descriptive study and the observed differences in clinical phenotypes (both for atherosclerosis as well as CHS) are mostly expected based on previous publications by the authors and others. Moreover, the authors try to “explain” their observations in the ApoE^{-/-} model with “similar” observations in CHS, although both disease models are fundamentally different in terms of mechanism of disease onset/initiation and progression. On the other hand, the intriguing finding that disrupted CD40L-CD40-mediated T cell-DC crosstalk simultaneously leads to impaired Treg- and Th1-differentiation remains unexplored. Nor is it investigated why/how an apparently reduced Treg immune regulation enables/leads to an attenuated Th1 effector response/inflammatory reaction. Similarly, analysis of the compromised B cell development and slightly reduced antibody titers in the absence of T cell CD40L also remain purely descriptive and lack any mechanistic insight.

Specific comments:

1- Results, line 142-144: “Accordingly, transcripts of several pro-inflammatory cytokines including interleukin (IL)-6 (p=0.01, Mann-Whitney test) and interferon (IFN)- γ (p=0.03, Mann-Whitney test) were decreased in the descending aorta. These results were phenocopied in DC-specific CD40 knockout mice (supplemental Figure S3a-e)...” – I cannot find any cytokine data. It would be more convincing to demonstrate elevated IL-6 and, in particular, IFN γ in the plaques, i.e. co-localization with CD4 by immunofluorescence microscopy (Fig. 1c, d) or FACS.

2- Fig.3: Please provide absolute cell numbers in a, d, and e. Consider to expand the y-axis in d; the differences in EM cells are barely visible. One can argue whether they are “profound” (line

176), but are they biologically significant?

3- Results, line 172-174: "...counterintuitively exhibited a reduced and more stable plaque phenotype." – Why is this "counterintuitive"? And how do you know that the plaques are "more stable"? On the other hand, what is more counterintuitive is why do these mice exhibit attenuated Th1 responses in the presence of fewer Tregs? Shouldn't the latter lead to compromised immune regulation and hence exaggerated Th1 reactivity? Please explain and address experimentally.

4- Fig.4: RNAseq analysis of total splenic CD4+ T cells represents an inconclusive experiment, resulting in a circular argument. Since both Treg and Th1 cells are diminished among the CD4 T cells (as shown in Fig.3), obviously their signature genes (a, FoxP3 and CXCR3, respectively) as well as associated signaling pathways (b, IL-2/STAT5 and IFNg) will be reduced as well. The authors should compare WT and CD40L-deficient T cells either by single cell RNA sequencing or repeat the experiment with, respectively, purified Treg and Th1 populations to unravel cell- or subpopulation-specific differences in gene expression and signaling pathways. And do Treg and Th1 cells from DC-specific CD40 KO mice differ or exhibit a similar phenotype than their CD40L-deficient counterparts?

5- Results, line 193-197: "...the profound decrease in the pro-inflammatory Th1 population is the main mechanism for the anti-inflammatory phenotype observed in Cd40lfl/fl/Cd4CreTreg mice while the decrease in Tregs may be attributed to either a developmental defect or a reduction in the immune system's requirement for suppressive Tregs under atherosclerotic conditions." – (1) The authors provide no evidence to call this an "anti-inflammatory phenotype", as it stands it is simply less pro-inflammatory. Moreover, fewer Th1 cells do not represent "the main mechanism" but merely a description of the phenotype (which is further complicated by the simultaneous drop in Tregs, which should facilitate enhanced Th1 inflammation). (2) The authors do not provide any data on Treg development, induction or differentiation in the absence of CD40L and hence the second part of the statement on "the decrease in Tregs" is pure speculation.

6- Fig.5: Similar to the atherosclerosis experiments, analysis of the CHS response remains descriptive. Moreover, it is difficult to understand how a skin contact allergy reaction triggered by different mechanisms should be helpful in explaining an atherosclerosis phenotype. However, similar to the atherosclerosis phenotype, how do the authors explain a 50% diminished ear swelling reaction (b) in the presence of 50% less Tregs (d), marginally reduced CD62Lneg effector cells (d), a similar frequency of LN DC (e) and only slightly reduced OX40L expression (f)? This requires a more in-depth analysis. To determine the hapten-specific activation of peripheral lymph node cells, i.e. DC and T cells, the authors may want to use the water-soluble form of the sensitizing hapten DNFB (DNBS). What is the pro- and anti-inflammatory cytokine profile of DC (subsets) both in T cell-specific CD40L KO and in DC-specific CD40 KO mice? Fig5e, f: Analysis of bulk CD11c+MHCII+ cells is uninformative. The authors should discriminate migratory from resident DC as well as distinct skin and lymph node DC subsets in their analysis. How about other co-stimulatory molecules like CD80/86 and CCR7? Please provide absolute cell numbers in b, d, and e.

7- Fig.6: As for the other experiments, analysis of the B cell phenotype remains completely descriptive. Please provide absolute cell numbers in a and b. Fig.6e, f: Are the mostly very small differences (on a linear scale) in Ab concentrations/titers biologically significant?

8- Discussion, line 256-264: "...IFNg may have allowed for decreased leukocyte infiltration into the plaque as well as an increase in SMC proliferation and collagen production. Thus, T cell CD40L primes atherosclerosis through IFN- γ activation, which helps to initiate both immune cell recruitment and fibrous cap thinning." – This needs to be demonstrated experimentally. Currently, this conclusion is not supported by the data.

Minor points:

9- Introduction, line 81-83: "...and severe thrombotic events as a result of disrupted CD40L-aIIbb3 interactions in arterial thrombi." – Please explain to the non-expert reader.

10- Results, line 224: "(Figure 5f-g)" – There is no figure g.

11- The authors should somehow incorporate the fact that all tg mice are on an ApoE-deficient background into the names of the different mouse lines. As it stands the tg mice (Cd40lfl/fl/Cd4Cretg, Cd40lfl/fl/Pf4Cretg and Cd40lfl/fl/Cd11cCretg) can be easily confused with conventional conditional knockouts.

Reviewer #4 (Remarks to the Author):

Lace et al investigate the role of CD40L in atherosclerosis and its cell-specific effects in leukocytes and platelets. They generated ApoE-deficient mice with conditional CD40L deletion in T cells and platelets. Whereas CD40L seems to mediate atherosclerosis in T cells, it does not seem to do so in platelets but there rathermore seems to facilitate atherothrombosis. The authors postulate that the role of CD40L in T cells on atherosclerosis is mediated is by reduced Th1 polarization, impaired antigen-dependent T cell proliferation and reduced oxLDL IgG production. CD40 deletion in dendritic cells seem to have similar effects. The authors draw a somehow vague conclusions: 'Together, our results illuminate the divergent cell-specific mechanisms of CD40-CD40L signaling in the various stages of atherosclerosis, which may lead to advances in targeted therapies'.

This is a well performed study and the data are well presented. The study is done by experts in the field and the manuscript is well written. The scope of the data is impressive.

The major limitation of the current study is the difficulty to see a clear-cut take home message. What is the actual mechanism by which CD40L impairs atherosclerosis? The authors should include a schematic drawing to help understand the conclusions and mechanisms proposed.

What is the actual evidence that CD40 is involved in the effects described by the authors? CD40 is stated very centrally in the title. The authors should explain how this is supported by their data.

The literature on the role of platelets in atherosclerosis in contrast to atherothrombosis should be critically discussed. The authors data on the CD40L platelet-specific deficiency is not supporting the role of platelets in atherothrombosis. Preclinical data as well as clinical data on the role of platelets on atherosclerosis should be critically discussed. The authors argument that only activated platelets play a role in atherosclerosis is not very convincing. Would this have any relevance for in vivo preclinical models or diseases in patients?

The authors speculate on cell selective inhibition of CD40L. Are there convincing therapeutic concepts in this regards? The authors should at least mention these.

The authors cite literature that CD40L is a risk marker for various diseases, including cardiovascular events. However, it seems the authors are quite biased in their selection of literature. There are also reports that CD40L is not suitable or a strong biomarker. The authors should discuss this in a more unbiased way.

The authors should discuss the potential interaction of CD40L with ligands of CD40L others than CD40.

Figure 1: The scope of investigation of atherosclerosis is quite limited. Collagen content, necrotic core size, abundance of various leukocyte subsets and other measures normally assessed for atherosclerosis would be of interest. The authors should also explain why they used chow diet and not high fat diet to investigate atherosclerosis in the ApoE^{-/-} mouse model.

Figure 2: These results are not really novel. The authors should clearly state this. Can the authors provide platelet staining in the injured arteries? Can the authors show that CD40L deficiency in T cells does not influence atherothrombosis?

Figure 6d: The stainings are difficult to assess. Can the authors provide a quantification? How many experiments are done here? IgM levels have been described as being protective in atherosclerosis. How does that fit to the authors' results? Also the literature on anti-oxLDL IgG is somehow controversial which needs further discussions.

Response to Reviewers' comments:**Reviewer #1:**

Dr. Lutgens and colleagues have published numerous reports on the role of CD40-CD154 interactions in a mouse model of atherosclerosis. In this latest report, they focus on source requirements for CD154 in driving disease in this model. Overall, the work presents new data addressing the long-standing controversy regarding the physiologic role of platelet vs. T cell-expressed CD154 in stimulating CD40 signaling in various scenarios. However, some aspects of the work are not well-presented and/or discussed, and there is a tendency throughout to over-interpret results.

We thank reviewer #1 for their valuable comments. We agree that parts of our work could have been better represented. We have carefully revised our previous manuscript, added novel experiments to strengthen our findings (see below), and we have rewritten large parts of the original manuscript.

In several regards, the Introduction is confusing and not entirely accurate. In the final paragraph of p. 3, Ref. 17 is focused upon TRAF6, which is downstream of CD40 signaling, not a 'signaling intermediate' of CD154. It's important to be specific about when the authors are discussing CD154 as a ligand for the signaling receptor CD40, rather than signaling to the cell via CD154 itself (which has been suggested, although it's controversial). Here, the authors seem to be suggesting that CD154 itself is an important signal transducer in atherosclerosis, but the signals in this study all seem to be sent through CD40, not CD154.

We have adapted the introduction accordingly. The introduction has now been rewritten from a cardiovascular perspective and we have indeed focused on CD40L as ligand for the signaling receptor CD40.

Revised manuscript pages: 3-4, Lines 1-44

And CD40 'signaling intermediates' are indeed well-characterized, in contradiction to the statements in lines 90-92, which are confusing. The statement in lines 97-98 is also misleading. It is well-established that CD40 signaling cooperates with BCR signaling, but CD40 does not itself 'stimulate' either BCR or TCR signaling.

We agree with this reviewer, and we have removed this statement.

In line 100, the phrase "pro-atherogenic subsets, including....." implies that >1 T cell subset has been implicated in both pro and anti-atherogenic activity, but only one subset is listed for each. If there are others, list them; if not, remove the plural form and the 'Including'.

We agree with this reviewer, and we have removed this statement.

The sentence in lines 104-106 was very awkward and confusing. Please re-word this.

We agree with this reviewer, and we have reworded this statement for clarity.

Revised manuscript page: 3, Lines 30-31

'Platelet CD40L, on the other hand, mainly signals through the integrin $\alpha\text{IIb}\beta\text{3}$ leading to platelet aggregation and endothelial activation^{26,27}.'

Finally, the Introduction barely mentions the long-standing controversy regarding the true physiologic role(s) of platelet CD154, which is a key topic addressed in this study. This should be discussed in the Introduction.

We have expanded the introduction and discussion sections in the revised manuscript to include more background on platelet CD40L and its role in atherosclerosis:

Introduction (pages 4, Lines 30-42):

'Platelet CD40L, on the other hand, mainly signals through the integrin $\alpha\text{IIb}\beta\text{3}$ leading to platelet aggregation and endothelial activation.'

Until now, it is unclear which CD40L expressing cell type(s) are responsible for the different aspects of atherogenesis. In the past, CD40L^{-/-} bone marrow chimeras did not show any effect on atherosclerosis, suggesting that CD40L-expressing non-hematopoietic cells, i.e., endothelial cells or VSMCs, drive atherosclerosis. On the other hand, transfer of thrombin-activated CD40L-deficient platelets ameliorated platelet-induced aggravation of atherosclerosis by preventing platelet-leukocyte aggregate formation as well as leukocyte recruitment, suggesting a major role for platelet CD40L in atherosclerosis. However, both approaches are rather unspecific: in a bone marrow transplantation, all hematopoietic lineages are transplanted, and the sub-lethal irradiation, required for the procedure, induces damage to the bone marrow stroma, which may impact mature immune cell function. Transfer of thrombin-activated CD40L^{-/-} platelets, only reflects the effects of platelet CD40L after strong activation, and although informative, does not delineate the role of platelet CD40L in atherogenesis.'

Discussion (pages 11-12, Lines 234-256):

'Platelets are the other major CD40L expressing cell type, and they represent an important player in atherosclerosis. Not only are platelets considered to play a role in the development of atherosclerosis, i.e. in the recruitment of immune cells to the arterial wall and atherosclerotic lesion, they also play a pivotal role in atherothrombosis: intravascular damage caused by the rupture of an atherosclerotic plaque, which leads to massive occlusive platelet aggregation thereby causing tissue ischemia, clinically manifested as myocardial infarction or stroke. Platelet CD40L has been reported to be involved in several immunological aspects of platelet function. In platelets, CD40L can signal via its receptor CD40, and activate NFκB signaling via TRAF2, thereby contributing to platelet activation and release of inflammatory chemokines. Platelet CD40L facilitates leukocyte recruitment to the arterial wall, the formation of platelet-leukocyte aggregates and platelet chemokine secretion, all mechanisms related to atherosclerosis development. However, the CD40-TRAF2 pathway only reflects a minor part of the CD40L cascade in platelets, and the interaction between CD40L and the integrin αIIbβ3 appeared to be more relevant. Deficiency of CD40L-αIIbβ3 interactions resulted in less αIIbβ3 activation, decreased thrombus growth, and increased thrombus stability, as also observed in the platelet-specific (Cd40l^{fl/fl}/Pf4Cre) model in our study, which appeared independent of the receptor CD40. In the present study, we did not observe any effect of selective deficiency of platelet CD40L on atherosclerotic plaque burden or phenotype. However, we did find a beneficial role for deficiency of platelet CD40L in an atherothrombosis model, confirming the importance of platelet CD40L in thrombus formation and stability. At first sight these findings seem contradictory to a previous study, where we demonstrated that repeated injection of thrombin activated CD40L-deficient platelets reduced atherosclerosis when compared to injection of thrombin activated wild type platelets by decreasing leukocyte recruitment, PLAs and thrombus formation. However, these results actually show that platelet CD40L affects atherogenesis in models causing a certain level of platelet activation.'

1. The mouse models used here have not all been referenced or adequately described. What was the source of CD11cCre-Apoe^{-/-}, Cd4Cre-Apoe^{-/-} and Pf4Cre-Apoe^{-/-} mice that were backcrossed to the CD40^{fl/fl} and CD154^{fl/fl} mice that the authors had produced for them? Either cite references in which those mice were characterized, or present data on their validation here. Importantly, none of the data presented show the level of normal CD154 expression on mouse platelets, nor the effectiveness of removal of this expression – only data on T cell and DC-specific knockout mice are presented. It is critical to also validate the platelet-specific knockout mouse, to accurately interpret results in this report.

We agree with the reviewer that the models were not sufficiently described or confirmed. Therefore, for each Cre model, we have now included references to the stock numbers from Jackson Laboratories as well as cited the original papers confirming their action.

Additionally, we have confirmed our platelet CD40L model using an ELISA to detect platelet-derived soluble CD40L (Supplemental Figure 1d). Platelet-specific deletion of CD40L led to undetectable plasma levels of soluble CD40L.

Revised manuscript page: 13, Lines 274-285

2. Although presented as statistically significant, some of the differences shown in the Figures are not biologically convincing and are over-interpreted. For example, the differences in Fig. 3d are barely detectable, but described as a 'profound reduction' in EM T cells on p. 7. In this Figure, actual cell numbers – not just percentages – should also be shown. Lines 182-184 of this page describe a 77% reduction in Th1-produced IFN-γ, but I could not find these data anywhere in the Figures. Either present the data or remove

their discussion from the text. Presumably the important site of action of T cells in this model is in the vasculature, not the spleen or LN – this needs to be considered when interpreting these data.

We agree that the figures may not have been biologically convincing for the effector cells in Figure 3d due to their presentation in the graph. Therefore, we have excluded the values for naïve T cells and central memory T cells as they remained unchanged between wild type and knockout mice (now Figure 3a). This figure has been updated to show a 38% reduction in EM T cells in the blood, 42% reduction in the EM T cells in the lymph node, and 37% reduction in the spleen of which all values are both statistically and biologically significant. In terms of absolute numbers, we can understand the reviewer's concern. However, we feel that absolute numbers do not normalize data between animals so differences in weight of the animal, weight of the organ, or amount of organ harvested is not accounted for. Instead, we have presented the percentage of the Tregs and EM T cells in reference to all white blood cells (Supplemental Figure 4b-c). Again, both T-cell subpopulations are significantly reduced across all organs, which points to a biological significance. According to proper animal welfare, we are not allowed to use additional animals when normalization against the percentage of white blood cells is a valid analysis.

Furthermore, we agree that the site of the vasculature is key for the reduction we see in atherosclerosis. We have expanded Figure 1 in our revised manuscript, which further characterizes the plaque phenotype of our T cell CD40L model. Aortic cytokine data, including the reduction in Th1-produced IFN- γ , is now reported within this figure (Figure 1h). We have also replicated this data in our dendritic cell CD40 model further cementing that the CD40-CD40L is a key modulator of Th1-produced IFN- γ (Supplemental Figure 3d).

We also added a human cohort of cerebrovascular disease and found that in both atherosclerotic plaques and in the circulation, sCD40L and sCD40 levels positively correlate with plaque and circulating IFN- γ concentrations

We have discussed these findings in the revised manuscript page 6, Lines 91-99:

'To investigate whether the CD40(L)-IFN- γ axis, observed in our in vivo mice models, is also present in patients, sCD40L and sCD40 levels were measured in atherosclerotic carotid plaques and plasma samples from 185 patients from the Carotid Plaque Imaging Project (CPIP) cohort (supplemental Table S2). Within the advanced carotid atherosclerotic plaques, both sCD40L ($r=0.264$, $p=0.000321$) and sCD40 ($r=0.211$, $p=0.004$) were positively correlated (Spearman correlation) with levels of IFN- γ in the plaque. Likewise, plasma sCD40L ($r=0.276$, $p=0.000113$) and sCD40 ($r=0.186$, $p=0.010$) concentrations also correlated (Spearman correlation) with circulating IFN- γ levels. These data suggest that, similarly to our hyperlipidemic mice, the CD40L-CD40 pathway also initiates a Th1-associated IFN- γ response in human atherosclerosis.'

3. Interpretation of data presented in Fig. 5 is confusing. How do the authors reconcile biologically decreased inflammation in 2 models in which there are fewer Treg?

We understand the reviewer's point of view; however, in both our T cell CD40L model (Figure 1h) and DC CD40 model (Supplemental Figure 3d) we have shown that the primary anti-inflammatory cytokines produced by Tregs, TGF- β and IL-10, are not reduced in the vasculature despite the reduced Treg numbers in both models. Together with the data obtained from the Treg suppression assay (Figure 4e), we believe the reduced numbers of Tregs are not involved in the reduction of atherosclerosis. Functionally, the Tregs appear to have similar suppressive capacity and cytokine expression suggesting the defect in their overall numbers might lie in their generation, as has been reported previously by Guiducci et al (2004).

Moreover, our animal model was based on a 28-week chow diet on an ApoE background to characterize late-stage atherosclerosis. Previous studies in atherosclerotic mouse models have revealed that Tregs are protective in early stages of atherosclerosis, whereas they only play a minor role in advanced atherosclerosis (reviewed in Spitz et al, 2016). In advanced atherosclerosis Tregs were reported to become FoxP3+IFN γ +, undermining their protective nature (Butcher et al, 2016).

In the revised manuscript, this has been elaborately discussed on pages 10-11, Lines 208-223-:

'Although Tregs are of major importance during atherogenesis, the reduction in Tregs that we observed in our T cell-deficient CD40L and DC-deficient CD40 mice did not result in aggravated atherosclerosis. On the contrary, blocking the T cell – DC CD40L-CD40 axis reduces atherosclerosis, revealing that a decreased Th1 response can overcome a reduction in immune-regulatory cells, most likely because less immune activation requires less immune suppression. While the reduced immune-suppressive requirement observed in our study may reduce the necessity for anti-inflammatory Tregs under atherosclerotic conditions, it is also important to note that T-cell subsets are highly plastic in terms of phenotypic characteristics during unresolving inflammation, especially in advanced stages of atherosclerosis which we have studied here. The number of Tregs was reported to be 3-times higher in early than in advanced stages of atherosclerosis⁴², indicating that Tregs exert most of their immune-regulatory effects in early atherosclerosis. Moreover, Tregs in more advanced stages of atherosclerosis exhibit signs of plasticity. They co-express Foxp3 and IFN- γ and show a mixed Th1/Th17/Treg based transcriptional profile, undermining their protective nature⁴³. We therefore assume that the decrease in Th1 response in our T cell-specific CD40L (Cd40^{fl/fl}/Cd4Cre^{tg}) deficient mice with advanced atherosclerosis affects atherogenesis to a larger extent than the decrease in, likely partly dysfunctional, Tregs.'

I don't follow the authors' conclusions about OX40L expression; the only data addressing this are in 5f, and I see no difference – listing a p value is not meaningful when the difference is undetectable. A statistical difference is not necessarily a biologically important difference. For both these data, and those mentioned above in Fig. 3, the discussion needs to acknowledge the magnitude of the differences seen, which is quite modest.

With regards to the contact hypersensitivity (CHS) experiment, we fully agree with the reviewer that this experiment is not very helpful to explain the role of T cell CD40L in atherosclerosis. We have therefore removed these data from our manuscript. However, in the revised version, we have replaced these data with a DC-T cell co-culture experiment based on the OVA-OTII system (Figure 5). Here we could strengthen the causal link between CD40 ligation and Th1 polarization through an in vitro T-cell assay utilizing CD40 deficient DCs together with naïve OT-II T cells. Similar to our in vivo results, deficiency of DC CD40 resulted in a lower frequency of effector T cells (Figure 5d) as well as lower interferon-gamma production (Figure 5e) upon OVA stimulation by the OT-II T cells. While interferon-gamma was significantly reduced in the culture supernatant, other T helper cytokines including IL-2, IL-4, IL-5, IL-10, IL-13, and IL-17A (Supplemental Figure 5b-c) remained unchanged with or without CD40 ligation reinforcing that the CD40-CD40L dyad primarily functions in Th1 polarization during co-stimulation.

4. Regarding data presented in Fig. 6, multiple published reports many years ago established a major defect in Ig isotype switching in the absence of CD154 (in both mice and people), so these results are hardly novel or surprising. The results do not establish that alterations in Ig production 'contributes' to the anti-inflammatory phenotype in the atherosclerosis model; this is a correlation, for which cause and effect is not tested here.

We agree with this reviewer's comment, and we added these data to confirm a hampered Ig isotype switching in our CD4Cre-CD40Lfl/fl-ApoE^{-/-} model. Our model indeed phenocopied the findings observed in the CD40L^{-/-} mouse model. Interestingly, the CD4Cre-CD40Lfl/fl-ApoE^{-/-} model not only exhibited a decrease in germinal center formation and total IgG levels, but also showed a specific decrease in anti-oxLDL IgG levels.

The potential impact of low total IgG levels and anti-oxLDL IgG levels on atherosclerosis, and, the potential contribution of this phenomenon (in addition to our main finding, the decreased Th1 response) to the observed decrease in atherosclerosis in the CD4Cre-CD40Lfl/fl-ApoE^{-/-} mice has been elaborately discussed in the revised manuscript page 11, Lines 224-233.

'Another phenotypic aspect of our model that, besides a decreased Th1 response, will have contributed to the observed decrease in atherosclerosis, is the reduction in IgG, and more specifically anti-oxLDL IgG antibodies. Germinal center derived IgG antibodies are known to promote atherogenesis and do this via multiple atherogenic pathways including metabolic reprogramming of immune cells, growth factors as well as immune cell recruitment⁴⁴. In addition, antibodies against oxidation specific epitopes, including oxidized (ox)LDL also play a major role in atherosclerosis⁴⁵. Whereas natural IgM oxLDL antibodies are considered to protect against atherosclerosis⁴⁶, via the induction of IL5, IgG antibodies against oxLDL are considered pro-atherogenic⁴⁷. The lack of Ig isotype switching, characteristic of CD40(L) deficiency, was also observed in our T cell deficient CD40L mice where total IgG, as well as oxLDL specific IgG, had decreased thereby contributing to the observed decrease in atherosclerosis.'

5. The same tendency to equate correlation with causality is found in lines 255-256 of the Discussion. The authors cannot claim that inhibiting the Th1 response driven by CD154 is the major mechanism by which atherosclerotic plaques are stabilized, when this causal relationship has not been tested; right now, it is a correlation.

We thank the reviewer for this insightful comment. To strengthen the causal relationship between CD40 ligation and Th1 polarization, we have now included an additional in vitro experiment testing the effect of dendritic cell CD40 ligation on T cell responses. Using bone marrow derived dendritic cells from CD40-expressing ApoE^{-/-} mice or ApoE^{-/-} mice with a global CD40 deficiency to present an OVA peptide to naïve OT-II T cells, we observed a reduction in activation of effector cells (Figure 5d) similar to our in vivo models. In addition, we tested the supernatant of this DC-T cell co-culture to characterize prototypical T cell-produced cytokines that may be affected by CD40 ligation. Most prominently, IFN- γ production was significantly reduced in the absence of dendritic cell CD40 (Figure 5e) while other T-cell cytokines including IL-2, IL-4, IL-5, IL-10, IL-13, and IL-17A remained unchanged (Supplemental Figure 5b-c).

6. What do the authors suggest is the source of the striking but anomalous Treg numbers, as mentioned above, which seem counter-intuitive with the anti-inflammatory effect? This should be addressed in the Discussion.

The reviewer is correct that our Treg phenotype is counter-intuitive; however, we believe our model of late-stage atherosclerosis is a very important consideration when analyzing the striking Treg numbers (see answer question 3). As we discussed in Question 3 above, most landmark papers describing the protective nature of Tregs used a shorter duration of diet, which likely produced earlier stages of atherosclerosis in their experiments. Butcher et al (2016) demonstrated a surprising plasticity in Treg phenotypes during unresolving inflammation, which might suggest a limited capacity to suppress inflammation in later stages of atherosclerosis.

The Discussion should also compare and contrast results obtained with the conditional knockout mice here with those obtained earlier by this group in the whole-mouse CD40-deficient knockout (e.g. Ref. 10).

We have now also added a paragraph to the introduction where we compare and contrast results obtained with our full knock-out models (revised manuscript page 4, lines 32-42):

'Until now, it is unclear which CD40L expressing cell type(s) are responsible for the different aspects of atherogenesis. In the past, CD40L^{-/-} bone marrow chimeras did not show any effect on atherosclerosis, suggesting that CD40L-expressing non-hematopoietic cells, i.e., endothelial cells or VSMCs, drive atherosclerosis^{28,29}. On the other hand, transfer of thrombin-activated CD40L-deficient platelets ameliorated platelet-induced aggravation of atherosclerosis by preventing platelet-leukocyte aggregate formation as well as leukocyte recruitment, suggesting a major role for platelet CD40L in atherosclerosis²⁷. However, both approaches are rather unspecific: in a bone marrow transplantation, all hematopoietic lineages are transplanted, and the sub-lethal irradiation, required for the procedure, induces damage to the bone marrow stroma, which may impact mature immune cell function. Transfer of thrombin-activated CD40L^{-/-} platelets, only reflects the effects of platelet CD40L after strong activation, and although informative, does not delineate the role of platelet CD40L in atherogenesis.'

Finally, the suggestion that the results presented here would allow cell-type-specific blocking of the CD40-CD154 interaction for therapy in vivo is baffling – how, exactly, do the authors envision this would be accomplished?

We agree with this reviewer that this suggestion was a bit over the top. We have now adjusted our final statement in the abstract and the discussion of the revised manuscript. However, we have included two potential therapeutic options which may allow for cell-specific targeting in the discussion.

Abstract

'Our results establish divergent and cell-specific roles of CD40L-CD40 in atherosclerosis, which has major implications for therapeutic strategies targeting this pathway.'

Discussion, Pages 11-12, Lines 266-270

'By deciphering the cell type-specific effects of the CD40L-CD40 pathway in disease, targeted therapeutic strategies directed against cell types (i.e., via cell-specific bi-specific antibodies⁵⁵ or targeted nanobiologics⁵⁶) or cell type-associated signaling intermediates, we will be able to more precisely target the desired effector pathways of the CD40L-CD40 dyad, thereby limiting unwanted side effects.'

Minor comment:

The authors use past vs. present tense seemingly randomly when describing previously-published work. I believe the convention is to use present tense for published data; past tense for unpublished results (such as the current report). Please be consistent throughout the manuscript.

We thank the reviewer for this helpful comment. We have revised the manuscript to reflect the correct verb tense.

Reviewer #2

This is an interesting and ambitious paper examining the role of CD40L-CD40 axis in atherosclerosis. Disruption of CD40 axis in T cells and in dendritic cells likely decreased atherosclerosis while disruption of platelet axis had no effect. These are interesting observations but lack of crucial experimental details does not allow confidence in the reported results at this stage. I would want to see convincing evidence of validation of the experimental models before accepting these results—in particular because some of the conclusions are at odds with existing paradigms—for example: the authors report that diminished Treg numbers are associated with decreased lesions—which the authors themselves recognize is counterintuitive. (Of course, other aspects are changed and these observations may be valid in context—but need to be sure models are appropriate).

We thank reviewer #2 for their useful comments. In the revised version of this manuscript, we have added experimental details and validation experiments confirming the appropriateness of our mouse models, and we have elaborated and provided an explanation on the observation reduced Treg numbers in spite of lower atherosclerotic burden (see below for specific details).

Major comments:

The two most important methodological concerns are the genetic background of the mice and the adequacy of the various gene knockdowns in these models.

First: It is well known in the atherosclerosis field that the genetic background of the mice can impact atherogenesis and that all experimental models must be on similar homogenous genetic backgrounds—which ideally should be at least 10 generations, but at minimum at least 7. There is no description of the genetic background of the floxed mice, the Cre mice and no indication of how many generations were these mice crossed to the apoE mice.—eg for the CD40L^{fl/fl} and the CD40^{fl/fl}—what were there genetic backgrounds vs the apoE—how many generations were these backcrossed to the apoE? Similarly—what were the genetic backgrounds of the CD11cCre and the CD4Cre and Pf4Cre? These were said to be on apoE background but how many generations were each?

We agree that the mouse models were not adequately described in the original submission. Therefore, we have included both the stock numbers of the Cre models from the Jackson Laboratory as well as citations where the models have been validated. Our CD40L^{fl/fl} and CD40^{fl/fl} models were generated on a C57Bl6J background. They were backcrossed to CD4-Cre and PF4-Cre mice that were also on a 100% C57Bl6J background. Additionally, we agree the ApoE background is an important factor in any atherosclerosis study. Therefore, all our mouse models used in this study were backcrossed >10x on the ApoE^{-/-} background. As you can imagine, this took quite some time and the generation of these mice was initiated in 2010. We have incorporated this information in the revised manuscript (revised manuscript, Page 13, Lines 274-285):

'For the atherosclerosis studies, Cd40^{fl/fl} mice were successfully generated via insertion of loxP sites flanking exon 3, which enabled cre-mediated deletion of exon 3 causing a translational frameshift rendering downstream exons nonfunctional (Ozgene Pty Ltd, Bentley, Australia, supplement Figure 1a)⁵⁷. Cd40^{fl/fl} mice were successfully generated via insertion of loxP sites flanking exons 2 and 3, which enabled cre-mediated deletion of exon 2 and 3 (Ozgene Pty Ltd, Bentley, Australia, supplement Figure 2a)⁵⁸. Cell-specific knockouts were generated by backcrossing Cd40^{fl/fl} and Cd40^{fl/fl} mice to either CD11cCre (stock No: 008068, Jackson Laboratory, Bar Harbor, Maine, USA)⁵⁹, Cd4Cre (stock No: 017336, Jackson Laboratory, Bar Harbor, Maine, USA)⁶⁰ or Pf4Cre (stock No: 008535, Jackson Laboratory, Bar Harbor, Maine, USA)⁶¹ transgenic mice. Finally, all strains were backcrossed at least ten times to ApoE^{-/-} transgenic mice (stock No. 002052, Jackson Laboratory, Bar Harbor, ME, USA) to generate atherosclerotic dendritic cell specific CD40 deficient mice or T cell and platelet specific CD40L deficient transgenic mice.'

Second: The extent of the various knockdowns is not at all clear? For example—how does Suppl Fig 1b and c show knockdown of CD40L in T cells—fig b is a functional assay of cells in artery and the fig c seems to show # (or mfi) of CD4 or CD8 T cells vs B—how does this show deletion or knockdown of CD40L expression in respective T cells—these cell types should be isolated and extent of knockdown (eg residual gene expression) documented directly to demonstrate efficiency of knockdown to be convincing.—The CD40 kd in Suppl fig 2 should be studied in similar manner—the data shown only speaks to relative cell # as I understand. Finally—I don't see any data supporting the validation of the platelet specific deletion of Cd40L.

We have included two new experiments to determine the % knockdown for the T cell and platelet CD40L knockout models. In Supplemental Figure 1b, we have isolated splenic, CD4+ T cells from wild type and knockout animals in the CD40LCD4Cre model. These cells were stimulated overnight with concanavalin A and cell-surface CD40L protein expression was measured by flow cytometry. Next in Supplemental Figure 1c, we have included a functional assay to measure soluble CD40L in the plasma of wild type and knockout animals in the CD40LPf4cre model, which revealed that sCD40L levels in PF4-CD40L mice were undetectable. While we agree that gene expression is a valid way to confirm the knockout models, we believe cell surface protein expression is critical to these experiments as CD40-CD40L crosstalk happens on the cell surface. Flow cytometric analysis allows for determination of cell lineage by the usage of specific markers ultimately generating a very specific analysis of protein expression especially on immune cell subpopulations with cell counts too low to isolate high quality RNA (i.e. dendritic cells).

Minor:

Page 6—IL-6 and IFN transcripts were measured in aorta—what does this mean if athero was decreased—less cells—what was denominator? What do the statistics refer to. You likely measured “several” transcripts- what other cytokines did you measure. Were others not changed—this information would be informative.

We agree with the reviewer that an overview of different aortic transcripts would help support the paper. Therefore, we have now included analysis of aortic IL-1 β , IL-6, IFN- γ , and TNF- α as prototypical pro-inflammatory cytokines together with TGF- β and IL-10 as prototypical anti-inflammatory cytokines (Figure 1h). Additionally, we have also included transcripts of CD3 and CD69, which are key T-cell markers. As the reviewer noted, only IL-6 and IFN- γ are changed in the T cell CD40L knockout mice suggesting lower expression of those cytokines in terms of all cells in the tissue. As IFN- γ is predominately produced by Th1 cells, a reduction in IFN- γ likely suggests a reduction in Th1 cells within the vasculature. The reduction in IL-6 suggests a lower inflammatory state. In terms of the analysis, RNA was isolated from atherosclerotic vasculature (ie. the descending aorta), and we normalized our data by using 100 ng/ul of RNA per sample before cDNA synthesis. 18s rRNA was used as a housekeeping gene to compare differential gene expression.

The reduction in total IgG2b is interesting, consistent with reduced T help, but why do you think IgE and IgM decreased? The role of IgG to OxLDL in atherogenesis is controversial but IgM to OxLDL is most likely protective and due to innate and not adaptive (at least directly) mechanisms, perhaps consistent that despite disruption of T cell help, IgM to OxLDL did not change.

We added these data to confirm a hampered Ig isotype switching in our CD4Cre-CD40L Δ -ApoE Δ model, as was observed previously in the full CD40L Δ model. CD40L-CD40 interactions are crucial for Ig isotype switching, so not only the switch from IgM to IgG, but also the switch to IgE (Fuleihan et al, 1993). Moreover, CD40L-CD40 interactions also are crucial for Ig production in general.

In the context of atherosclerosis, our CD4Cre-CD40L Δ -ApoE Δ model not only exhibited a decrease in germinal center formation and total IgG levels, but also showed a specific decrease in anti-oxLDL IgG levels. The potential impact of low total IgG levels and anti-oxLDL IgG levels on atherosclerosis, and the potential contribution of this phenomenon (in addition to our main finding, the decreased Th1 response) to the observed decrease in atherosclerosis in the CD4Cre-CD40L Δ -ApoE Δ mice has been elaborately discussed in the revised manuscript page 11, Lines 224-233.

‘Another phenotypic aspect of our model that, besides a decreased Th1 response, will have contributed to the observed decrease in atherosclerosis, is the reduction in IgG, and more specifically anti-oxLDL IgG antibodies. Germinal center derived IgG antibodies are known to promote atherogenesis and do this via multiple atherogenic pathways including metabolic reprogramming of immune cells, growth factors as well as immune cell recruitment⁴⁴. In addition, antibodies against oxidation specific epitopes, including oxidized (ox)LDL also play a major role in atherosclerosis⁴⁵. Whereas natural IgM oxLDL antibodies are considered to protect against atherosclerosis⁴⁶, via the induction of IL5, IgG antibodies against oxLDL are considered pro-atherogenic⁴⁷. The lack of Ig isotype switching, characteristic of CD40(L) deficiency, was also observed in our T cell deficient CD40L mice where total IgG, as well as oxLDL specific IgG, had decreased thereby contributing to the observed decrease in atherosclerosis.’

Table 1: The chol levels shown in Table 1 appear to be wrong units 3-4 mg/dL (? mmol/L). These also very low levels for apoE mice and too low to cause extensive atherogenesis. The chol levels of each group at the end of the 28 diet period should be shown—and ideally at the beginning. As these are apoE mice—the TG levels should be shown

We thank the reviewer for pointing out that we have listed the wrong unit for cholesterol levels. We have now changed Supplemental Table 1 to reflect the correct unit, i.e. mmol/L. While the cholesterol levels of these mice are low, it is important to note they did not receive a high fat diet. The mice were aged for 6 months (28 weeks) using a chow diet to study late stage atherosclerosis. Considering we did not implement a high fat diet, the cholesterol levels before the experiment were not measured as it is expected that they should be similar at the end of the aging experiment. TG levels were not measured as we did not fast our mice before sacrifice. Considering the ApoE background of all mice used in this study, they will spontaneously generate atherosclerotic plaques in their vasculature. Accordingly, six months of aging will induce large plaque growth as observed in all models.

General comment: While these observations may help explain the pathophysiology of atherogenesis and the possible complex roles of CD40L-CD40 interactions, at a practical level they do not likely offer specific targets for therapeutic interventions in the context of altering atherogenesis due to their impact on other immunological processes and autoimmunity. My own suggestion would be to refocus the Discussion on how knowledge of these interactions in atherosclerosis might help inform the impact of interfering on this pathway in other diseases.

We fully agree with reviewer #2, and we have fully rewritten the abstract's conclusion and the discussion section:

Abstract

'Our results establish divergent and cell-specific roles of CD40L-CD40 in atherosclerosis, which has major implications for therapeutic strategies targeting this pathway.'

Discussion, Pages 11-12, Lines 266-270

'By deciphering the cell type-specific effects of the CD40L-CD40 pathway in disease, targeted therapeutic strategies directed against cell types (i.e., via cell-specific bi-specific antibodies⁵⁵ or targeted nanobiologics⁵⁶) or cell type-associated signaling intermediates, we will be able to more precisely target the desired effector pathways of the CD40L-CD40 dyad, thereby limiting unwanted side effects.'

Reviewer #3

In this study, Lacy et al. address the role of the CD40-CD40L axis in atherosclerosis. To this aim, the authors have generated T cell- and platelet-specific CD40L-deficient and DC-specific CD40-deficient mice, and backcrossed these conditional mutants onto an atherosclerosis prone Apoe^{-/-} background. At 28 weeks of age, both T cell-specific CD40L and DC-specific CD40 deficiency reduced plaque progression, associated with hampered Th1 polarization, impaired antigen-dependent T cell proliferation and reduced oxLDL IgG production. In contrast, platelet-specific CD40L deficiency failed to decrease atherosclerosis, but ameliorated atherothrombosis. Altogether, this work represents a largely descriptive study and the observed differences in clinical phenotypes are mostly expected based on previous publications by the authors and others.

We thank reviewer #3 for their useful comments. We respectfully disagree that the 'differences in clinical phenotypes' are 'largely expected'. As reviewer #2 already points out, there are many contrasting results on the impact of the different CD40L and CD40-expressing cell types on the diverse aspects of atherosclerosis initiation and progression. To stress the importance of the current study's findings, we have now added a paragraph to the discussion where we compare, and contrast results obtained with our full knock-out models (revised manuscript, page 4, Lines 32-42):

'Until now, it is unclear which CD40L expressing cell type(s) are responsible for the different aspects of atherogenesis. In the past, CD40L^{-/-} bone marrow chimeras did not show any effect on atherosclerosis, suggesting that CD40L-expressing non-hematopoietic cells, i.e., endothelial cells or VSMCs, drive atherosclerosis^{28,29}. On the other hand, transfer of thrombin-activated CD40L-deficient platelets ameliorated platelet-induced aggravation of atherosclerosis by preventing platelet-leukocyte aggregate formation as well as leukocyte recruitment, suggesting a major role for platelet CD40L in atherosclerosis²⁷. However, both approaches are rather unspecific: in a bone marrow transplantation, all hematopoietic lineages are transplanted, and the sub-lethal irradiation, required for the procedure, induces damage to the bone marrow stroma, which may impact mature immune cell function. Transfer of thrombin-activated CD40L^{-/-} platelets, only reflects the effects of platelet CD40L after strong activation, and although informative, does not delineate the role of platelet CD40L in atherogenesis.'

Moreover, the authors try to "explain" their observations in the Apoe^{-/-} model with "similar" observations in CHS, although both disease models are fundamentally different in terms of mechanism of disease onset/initiation and progression.

We fully agree with reviewer #3 that the CHS model does not reflect the events during atherogenesis, and we have therefore removed these data from the manuscript.

On the other hand, the intriguing finding that disrupted CD40L-CD40-mediated T cell-DC crosstalk simultaneously leads to impaired Treg- and Th1-differentiation remains unexplored. Nor is it investigated why/how an apparently reduced Treg immune regulation enables/leads to an attenuated Th1 effector response/inflammatory reaction. Similarly, analysis of the compromised B cell development and slightly reduced antibody titers in the absence of T cell CD40L also remain purely descriptive and lack any mechanistic insight.

In the revised version of this manuscript, we have addressed these findings in more detail.

Specific comments:

1- Results, line 142-144: "Accordingly, transcripts of several pro-inflammatory cytokines including interleukin (IL)-6 (p=0.01, Mann-Whitney test) and interferon (IFN)- γ (p=0.03, Mann-Whitney test) were decreased in the descending aorta. These results were phenocopied in DC-specific CD40 knockout mice (supplemental Figure S3a-e)..." – I cannot find any cytokine data. It would be more convincing to demonstrate elevated IL-6 and, in particular, IFN γ in the plaques, i.e. co-localization with CD4 by immunofluorescence microscopy (Fig.1c, d) or FACS.

We agree that the site of the vasculature is key for the reduction in cytokines that we observe. We have extended Figure 1 to further characterize the plaque phenotype of our T-cell model, which also includes the aortic cytokine data (Figure 1h). We would also like to point out that this data is confirmed in our dendritic

cell CD40 model further cementing that the CD40-CD40L is a key modulator of Th1-produced IFN- γ (Supplemental Figure 3d).

2- Fig.3: Please provide absolute cell numbers in a, d, and e. Consider to expand the y-axis in d; the differences in EM cells are barely visible. One can argue whether they are “profound” (line 176), but are they biologically significant?

We agree that the figures may not have been biologically convincing for the effector cells in Figure 3d due to their presentation in the graph. Therefore, we have excluded the values for naïve T cells and central memory T cells as they remained unchanged between wild type and knockout mice. In our revised manuscript, Figure 3a has been updated to show a 38% reduction in EM T cells in the blood, 42% reduction in the EM T cells in the lymph node, and 37% reduction in the spleen of which all values are both statistically and biologically significant. In terms of absolute numbers, we can understand the reviewer’s concern. Considering WBC counts did not drastically differ between wild type and transgenic mice (Supplemental Table 1), we feel that absolute numbers may not give the reviewer the answer they are looking for as they do not normalize data between animals so differences in weight of the animal, weight of the organ, or amount of organ harvested is not accounted for. Instead, we have presented the percentage of the Tregs and EM T cells in reference to all white blood cells (Supplemental Figure 4b-c). Again, both T-cell subpopulations are significantly reduced across all organs, which points to a biological significance. According to proper animal welfare, we are not allowed to use additional animals when normalization against the percentage of white blood cells is a valid analysis.

3- Results, line 172-174: “...counterintuitively exhibited a reduced and more stable plaque phenotype.” – Why is this “counterintuitive”? And how do you know that the plaques are “more stable”? On the other hand, what is more counterintuitive is why do these mice exhibit attenuated Th1 responses in the presence of fewer Tregs? Shouldn’t the latter lead to compromised immune regulation and hence exaggerated Th1 reactivity? Please explain and address experimentally.

We fully agree with these reviewer’s comments. Indeed, the term counterintuitive, is not entirely correct, and we have explained this phenomenon in more detail in the revised version of our manuscript.

First, in the revised version of the manuscript, we more carefully phenotyped the atherosclerotic lesions as shown in Figure 1. In CD4tg-CD40fl-ApoE^{-/-} mice, atherosclerotic plaques were not only smaller in size, but also contained smaller necrotic cores (% wise), more VSMCs and thicker fibrous caps, which all resemble characteristics of a stable atherosclerotic plaque phenotype that is less prone to plaque rupture (Virmani et al, 2000).

Second, we understand the reviewer’s point about compromised immune regulation; however, in both our T cell CD40L model (Figure 1h) and DC CD40 model (Supplemental Figure 3d) we have shown that the primary anti-inflammatory cytokines produced by Tregs, TGF- β and IL-10, are not reduced in the vasculature despite the reduced Treg numbers in both models. Together with the data obtained from the Treg suppression assay (Figure 3e), we believe the reduced numbers of Tregs are not involved in the reduction of atherosclerosis. Functionally, the Tregs appear to have similar suppressive capacity and cytokine expression suggesting the defect in their overall numbers might lie in their generation, as has been reported previously by Guiducci et al (2004). Moreover, our animal model was based on a 28-week chow diet on an ApoE background to characterize late-stage atherosclerosis. Previous studies in atherosclerotic mouse models have revealed that Tregs are protective in early stages of atherosclerosis, whereas they only play a minor role in advanced atherosclerosis (reviewed in Spitz et al). In advanced atherosclerosis Tregs were reported to become FoxP3+IFN γ +, undermining their protective nature (Butcher et al, 2016).

In the revised manuscript, this has been elaborately discussed on pages 10-11, Lines 208-223 of the revised manuscript:

‘Although Tregs are of major importance during atherogenesis, the reduction in Tregs that we observed in our T cell-deficient CD40L and DC-deficient CD40 mice did not result in aggravated atherosclerosis. On the contrary, blocking the T cell – DC CD40L-CD40 axis reduces atherosclerosis, revealing that a decreased Th1 response can overcome a reduction in immune-regulatory cells, most likely because less immune activation requires less immune suppression. While the reduced immune-suppressive requirement observed in our study may reduce the necessity for anti-inflammatory Tregs under atherosclerotic conditions, it is also important to note that T-cell subsets are highly plastic in terms of phenotypic characteristics during unresolving inflammation, especially in advanced stages of atherosclerosis which we have studied here. The number of Tregs was reported to be 3-times higher in early than in advanced stages of

atherosclerosis⁴², indicating that Tregs exert most of their immune-regulatory effects in early atherosclerosis. Moreover, Tregs in more advanced stages of atherosclerosis exhibit signs of plasticity. They co-express Foxp3 and IFN- γ and show a mixed Th1/Th17/Treg based transcriptional profile, undermining their protective nature⁴³. We therefore assume that the decrease in Th1 response in our T cell-specific CD40L (Cd40^{fl/fl}/Cd4Cre^{tg}) deficient mice with advanced atherosclerosis affects atherogenesis to a larger extent than the decrease in, likely partly dysfunctional, Tregs.'

4- Fig.4: RNAseq analysis of total splenic CD4+ T cells represents an inconclusive experiment, resulting in a circular argument. Since both Treg and Th1 cells are diminished among the CD4 T cells (as shown in Fig.3), obviously their signature genes (a, FoxP3 and CXCR3, respectively) as well as associated signaling pathways (b, IL-2/STAT5 and IFN γ) will be reduced as well. The authors should compare WT and CD40L-deficient T cells either by single cell RNA sequencing or repeat the experiment with, respectively, purified Treg and Th1 populations to unravel cell- or subpopulation-specific differences in gene expression and signaling pathways. And do Treg and Th1 cells from DC-specific CD40 KO mice differ or exhibit a similar phenotype than their CD40L-deficient counterparts?

We can understand the reviewer's point of view; however, we respectfully disagree that gene expression will always and reliably predict protein expression especially in the case of immune cells like CD4+ T cells, which are known to rely on splice variants. Mechanistically, single cell sequencing primarily should answer questions of cell heterogeneity. Therefore, a single cell sequencing experiment would likely tell us that are differences in the Treg and Th1 populations between the two genotypes, which could be met with a reviewer suggesting our FACs data and single cell data are making a circular argument. Additionally, single cell sequencing of CD4+ subpopulations does not reliably produce clusters based on gene expression. In our personal experience as well as published reports (please see Cano-Gamez et al (2020), PMID: 32286271), single cell sequencing produces a gradient without clear borders on subpopulations meaning subpopulations of effector T cells would likely not be distinguishable and our reduced Th1 response would be hidden.

On the other hand, bulk sequencing of the same exact cell population would likely result in no differential gene expression between Th1 cells from CD40L wild type T cells and CD40L knockout cells. Logistically, purification of these two populations would require intracellular staining for key transcription factors and cytokines. This procedure would require the cells to be fixed and permeabilized, which would lead to significant damage and degradation of RNA. Therefore, we believe bulk sequencing of all CD4 cells (as shown in Figure 4) is the best experimental design to avoid these problems. The clear changes in Th1 and Treg expression markers is striking considering these populations were diluted in the total CD4 population that was sequenced. The ability to show changes in key cell lineage markers shows a robust change in both populations.

As it stands, we have validated the decrease in the Treg and Th1 subpopulations by two separate methods (ie. RNAseq and flow cytometry). Additionally, we have shown in the dendritic cell CD40 model a similar decrease in Treg and Th1 populations by flow cytometry (Supplemental Figure 3) likely suggesting the changes we have observed are due the absence of CD40-CD40L crosstalk between dendritic cells and T cells. Furthermore, our in vitro model of CD40 ligation (Figure 5) showed protein expression of IFN- γ was also significantly reduced while other T cell cytokines remained unaffected. Finally, we have also added a human cohort, which correlates both plaque and plasma levels of CD40 and CD40L to increased IFN- γ concentrations.

5- Results, line 193-197: "...the profound decrease in the pro-inflammatory Th1 population is the main mechanism for the anti-inflammatory phenotype observed in Cd40fl/fl/Cd4Cre^{tg} mice while the decrease in Tregs may be attributed to either a developmental defect or a reduction in the immune system's requirement for suppressive Tregs under atherosclerotic conditions." – (1) The authors provide no evidence to call this an "anti-inflammatory phenotype", as it stands it is simply less pro-inflammatory. Moreover, fewer Th1 cells do not represent "the main mechanism" but merely a description of the phenotype (which is further complicated by the simultaneous drop in Tregs, which should facilitate enhanced Th1 inflammation). (2) The authors do not provide any data on Treg development, induction or differentiation in the absence of CD40L and hence the second part of the statement on "the decrease in Tregs" is pure speculation.

We agree that this is a semantics issue in calling the phenotype 'anti-inflammatory' and a more appropriate description would indeed be less pro-inflammatory. Therefore, we have changed this description in the manuscript to match the correct assessment.

To strengthen the causal link between CD40 ligation and Th1 polarization, we have included an additional experiment testing the effect of CD40 ligation on T-cell polarization in vitro (Figure 5). In the absence of dendritic cell CD40, we observed a similar reduction in EM T cells (Figure 5d) as with our in vivo models. Furthermore, a lack of CD40 ligation significantly reduced T cell-produced IFN- γ (Figure 5e). We also tested additional T cell produced cytokines including IL-2, IL-4, IL-5, IL-10, IL-13, and IL-17A (Supplemental Figure 5b-c), but only IFN- γ was changed suggesting the CD40-CD40L acts predominately in Th1 polarization. When putting this data together with reduction of IFN- γ in both the T cell CD40L model (Figure 1h) as well as the dendritic cell CD40 model (Supplemental Figure 3d), we believe that this is strong evidence for the role of Th1-produced IFN- γ in the reduction of atherosclerosis in our in vivo models. Furthermore, we have characterized the plaque phenotype of our T cell CD40L model in depth in Figure 1. Our discussion puts these results into context with previous findings (revised manuscript, Page 10, Lines 192-199):

'The pro-inflammatory and pro-atherogenic nature of Th1 cells, along with their prototypical cytokine IFN- γ , is well documented². IFN- γ has been shown to activate and potentiate the expression of endothelial expressed adhesion molecules and pro-inflammatory chemokines allowing for the recruitment of leukocytes to the plaque³². Additionally, IFN- γ impairs SMC-mediated collagen synthesis thus weakening of the fibrous cap³³. Mice deficient in IFN- γ develop significantly smaller, more collagenous lesions resembling our phenotype³⁴. Thus, our data strongly suggest that T cell CD40L primes atherosclerosis through IFN- γ activation, which initiates immune cell recruitment, necrotic core formation, and fibrous cap thinning.

We also added a human cohort of cerebrovascular disease to validate our findings in humans and found that in both atherosclerotic plaques and in the circulation, sCD40L and sCD40 levels correlate with plaque and circulating IFN- γ concentrations, suggesting that the CD40L-CD40 pathway also initiates a Th1-associated IFN- γ response in human atherosclerosis (revised manuscript, Page 6, Lines 90-99):

'To investigate whether the CD40(L)-IFN- γ axis, observed in our in vivo mice models, is also present in patients, sCD40L and sCD40 levels were measured in atherosclerotic carotid plaques and plasma samples from 185 patients from the Carotid Plaque Imaging Project (CPIP) cohort (supplemental Table S2). Within the advanced carotid atherosclerotic plaques, both sCD40L ($r=0.264$, $p=0.000321$) and sCD40 ($r=0.211$, $p=0.004$) were positively correlated (Spearman correlation) with levels of IFN- γ in the plaque. Likewise, plasma sCD40L ($r=0.276$, $p=0.000113$) and sCD40 ($r=0.186$, $p=0.010$) concentrations also correlated (Spearman correlation) with circulating IFN- γ levels. These data suggest that, similarly to our hyperlipidemic mice, the CD40L-CD40 pathway also initiates a Th1-associated IFN- γ response in human atherosclerosis.'

With regards to the Tregs, we understand the reviewer's point of view; however, we believe that in our model, Tregs are of less importance and the decrease in Th1 responses (see answer to question 1). We are aware that deficiency of CD40L-CD40 interactions, as has been reported previously, causes a defective Treg development. Although this is not the focus of the present paper, we have elaborated on this phenomenon in the discussion (revised manuscript, Page 10, Lines 201-204):

'The involvement of CD40L-CD40 interactions in Treg development has been reported before in global CD40^{-/-} mice³⁵, and it was shown that that CD40 expression on DCs, thymic epithelial cells, and thymic B cells is critical for the in vivo generation and/or homeostasis of thymic Foxp3⁺ Tregs^{36,37}. However, how T cell CD40L deficiency contributes to Treg development exactly still needs to be elucidated.'

6- Fig.5: Similar to the atherosclerosis experiments, analysis of the CHS response remains descriptive. Moreover, it is difficult to understand how a skin contact allergy reaction triggered by different mechanisms should be helpful in explaining an atherosclerosis phenotype. However, similar to the atherosclerosis phenotype, how do the authors explain a 50% diminished ear swelling reaction (b) in the presence of 50% less Tregs (d), marginally reduced CD62Lneg effector cells (d), a similar frequency of LN DC (e) and only slightly reduced OX40L expression (f)? This requires a more in-depth analysis. To determine the hapten-specific activation of peripheral lymph node cells, i.e. DC and T cells, the authors may want to use the water-soluble form of the sensitizing hapten DNFB (DNBS). What is the pro- and anti-inflammatory cytokine profile of DC (subsets) both in T cell-specific CD40L KO and in DC-specific CD40 KO mice? Fig5e, f: Analysis of bulk CD11c+MHCII+ cells is uninformative. The authors should discriminate migratory from resident DC as well as distinct skin and lymph node DC subsets in their analysis. How about other co-stimulatory molecules like CD80/86 and CCR7? Please provide absolute cell numbers in b, d, and e.

We fully agree with reviewer #3 that the CHS model does not reflect the events during atherogenesis, and we have therefore removed these data from the manuscript.

7- Fig.6: As for the other experiments, analysis of the B cell phenotype remains completely descriptive. Please provide absolute cell numbers in a and b. Fig.6e, f: Are the mostly very small differences (on a linear scale) in Ab concentrations/titers biologically significant?

Considering this data is not the central point of our manuscript, we have moved this figure to the supplemental (Supplemental Figure 6). However, we believe it's necessary to include this data as it confirms a hampered Ig isotype switching in our CD4Cre-CD40Lfl/fl-ApoE^{-/-} model, as was observed previously in the full CD40L^{-/-} model. However, in the context of atherosclerosis, our CD4Cre-CD40Lfl/fl-ApoE^{-/-} model not only exhibited a decrease in germinal center formation and total IgG levels, but also showed a specific decrease in anti-oxLDL IgG levels. Therefore, this may have some effect on progression of atherosclerosis observed in our model. The potential impact of low total IgG levels and anti-oxLDL IgG levels on atherosclerosis, and the potential contribution of this phenomenon (in addition to our main finding, the decreased Th1 response) to the observed decrease in atherosclerosis in the CD4Cre-CD40Lfl/fl-ApoE^{-/-} mice has been elaborately discussed in the revised manuscript page 11, Lines 224-233:

'Another phenotypic aspect of our model that, besides a decreased Th1 response, will have contributed to the observed decrease in atherosclerosis, is the reduction in IgG, and more specifically anti-oxLDL IgG antibodies. Germinal center derived IgG antibodies are known to promote atherogenesis and do this via multiple atherogenic pathways including metabolic reprogramming of immune cells, growth factors as well as immune cell recruitment⁴⁴. In addition, antibodies against oxidation specific epitopes, including oxidized (ox)LDL also play a major role in atherosclerosis⁴⁵. Whereas natural IgM oxLDL antibodies are considered to protect against atherosclerosis⁴⁶, via the induction of IL5, IgG antibodies against oxLDL are considered pro-atherogenic⁴⁷. The lack of Ig isotype switching, characteristic of CD40(L) deficiency, was also observed in our T cell deficient CD40L mice where total IgG, as well as oxLDL specific IgG, had decreased thereby contributing to the observed decrease in atherosclerosis.'

8- Discussion, line 256-264: "...IFN γ may have allowed for decreased leukocyte infiltration into the plaque as well as an increase in SMC proliferation and collagen production. Thus, T cell CD40L primes atherosclerosis through IFN- γ activation, which helps to initiate both immune cell recruitment and fibrous cap thinning." – This needs to be demonstrated experimentally. Currently, this conclusion is not supported by the data.

We agree with the reviewer that our conclusions need additional support and have therefore performed additional experiments confirming the importance of the T cell CD40L- IFN- γ axis (see answers to comment #5). In this respect, we have also analyzed the atherosclerotic aorta in more detail. As already mentioned in the answers to comment #3, we have expanded Figure 1, which is dedicated to further characterizing the plaque phenotype observed in our T cell CD40L model. Within this figure, we describe a decrease in aortic IFN- γ expression, which parallels the decrease in T cell-produced IFN- γ we have observed in the absence of dendritic cell CD40 (Supplemental Figure 3d). Phenotypically, we observed a shift in plaque stages towards earlier stages (Figure 1d). Additionally, we observed an increase in smooth muscle cells within the plaque (Figure 1f) as well as an increase in fibrous cap thickness (Figure 1e) together with a decrease in necrotic core content (Figure 1g) of the plaque, which are all characteristic of more stable plaques.

Together with our existing data in Figure 1 (T cell, macrophage data), we believe our data mechanistically support the loss of CD40 ligation leading to reductions in IFN- γ in the vasculature. IFN- γ has been linked previously to destabilizing atherosclerotic plaques, which links our diminished IFN- γ concentrations to increased smooth muscle content.

Revised manuscript page 10, Lines 192-199:

'The pro-inflammatory and pro-atherogenic nature of Th1 cells, along with their prototypical cytokine IFN- γ , is well documented². IFN- γ has been shown to activate and potentiate the expression of endothelial expressed adhesion molecules and pro-inflammatory chemokines allowing for the recruitment of

leukocytes to the plaque³². Additionally, IFN- γ impairs SMC-mediated collagen synthesis thus weakening of the fibrous cap³³. Mice deficient in IFN- γ develop significantly smaller, more collagenous lesions resembling our phenotype³⁴. Thus, our data strongly suggest that T cell CD40L primes atherosclerosis through IFN- γ activation, which initiates immune cell recruitment, necrotic core formation, and fibrous cap thinning.

Minor points:

9- Introduction, line 81-83: "...and severe thrombotic events as a result of disrupted CD40L- α IIb β 3 interactions in arterial thrombi." – Please explain to the non-expert reader.

We are sorry for the confusion, and we have moved this part of the introduction to the discussion section.

Revised manuscript page 12, Line 240-248:

'In platelets, CD40L can signal via its receptor CD40, and activate NF κ B signaling via TRAF2, thereby contributing to platelet activation and release of inflammatory chemokines⁵⁰. Platelet CD40L facilitates leukocyte recruitment to the arterial wall, the formation of platelet-leukocyte aggregates and platelet chemokine secretion²⁷, all mechanisms related to atherosclerosis development. However, the CD40-TRAF2 pathway only reflects a minor part of the CD40L cascade in platelets, and the interaction between CD40L and the integrin α IIb β 3 appeared to be more relevant^{26,51}. Deficiency of CD40L- α IIb β 3 interactions resulted in less α IIb β 3 activation, decreased thrombus growth, and increased thrombus stability, as also observed in the platelet-specific (Cd40^{f^f/f^f}/Pf4Cre) model in our study, which appeared independent of the receptor CD40⁵¹.

10- Results, line 224: "(Figure 5f-g)" – There is no figure g.

We thank the reviewer for catching this oversight, which has been resolved now.

11- The authors should somehow incorporate the fact that all tg mice are on an ApoE-deficient background into the names of the different mouse lines. As it stands the tg mice (Cd40lf/f/Cd4Cretg, Cd40lf/f/Pf4Cretg and Cd40lf/f/Cd11cCretg) can be easily confused with conventional conditional knockouts.

We thank the reviewer for this comment. Considering the complicated nature of our paper with several models, we have purposely tried to simplify and color code the naming scheme to make it easier for the reader to differentiate the different models. Therefore, we have also decided to leave ApoE out of the naming scheme as it could add an additional layer for the reader to decipher. We have updated the methods section to reinforce that our mice were on an ApoE^{-/-} background (revised manuscript, page 13, Lines 282-285):

'Finally, all strains were backcrossed at least ten times to ApoE^{-/-} transgenic mice (stock No. 002052, Jackson Laboratory, Bar Harbor, ME, USA) to generate atherosclerotic dendritic cell specific CD40 deficient mice or T cell and platelet specific CD40L deficient transgenic mice.'

Reviewer #4

Lacey et al investigate the role of CD40L in atherosclerosis and its cell-specific effects in leukocytes and platelets. They generated ApoE-deficient mice with conditional CD40L deletion in T cells and platelets. Whereas CD40L seems to mediate atherosclerosis in T cells, it does not seem to do so in platelets but there rather more seems to facilitate atherothrombosis. The authors postulate that the role of CD40L in T cells on atherosclerosis is mediated is by reduced Th1 polarization, impaired antigen-dependent T cell proliferation and reduced oxLDL IgG production. CD40 deletion in dendritic cells seem to have similar effects. The authors draw a somehow vague conclusions: 'Together, our results illuminate the divergent cell-specific mechanisms of CD40-CD40L signaling in the various stages of atherosclerosis, which may lead to advances in targeted therapies'.

This is a well performed study and the data are well presented. The study is done by experts in the field and the manuscript is well written. The scope of the data is impressive.

We thank the reviewer for their comments, and we have more carefully formulated our conclusion in the revised version of our manuscript.

The major limitation of the current study is the difficulty to see a clear-cut take home message. What is the actual mechanism by which CD40L impairs atherosclerosis? The authors should include a schematic drawing to help understand the conclusions and mechanisms proposed.

We thank the reviewer for this helpful comment. We have created a graphical abstract to make the take home message clear to the reader.

What is the actual evidence that CD40 is involved in the effects described by the authors? CD40 is stated very centrally in the title. The authors should explain how this is supported by their data.

We understand the reviewer's concern, especially since CD40L can also signal via alternative receptors such as Mac1, but we believe that Supplemental Figure 3, where we show that a conditional dendritic cell CD40 knockout replicates the findings in our T cell CD40L knockout model, cements the role of CD40 ligation in IFN γ -induced plaque destabilization. Furthermore, we have added an additional in vitro experiment comparing T cell-produced cytokines with or without CD40 ligation from bone marrow derived dendritic cells (Figure 5). In the absence of CD40, OT-II T cells produced significantly less IFN γ while other T-cell cytokines including IL-2, IL-4, IL-5, IL-10, IL-13, and IL-17a remained unchanged.

The literature on the role of platelets in atherosclerosis in contrast to atherothrombosis should be critically discussed. The authors data on the CD40L platelet-specific deficiency is not supporting the role of platelets in atherothrombosis. Preclinical data as well as clinical data on the role of platelets on atherosclerosis should be critically discussed. The authors argument that only activated platelets play a role in atherosclerosis is not very convincing. Would this have any relevance for in vivo preclinical models or diseases in patients?

We thank the reviewer for their comment. In the revised manuscript, we discuss the role of platelets in atherosclerosis further (pages 10-11, Lines 235-256):

'Not only are platelets considered to play a role in the development of atherosclerosis, i.e. in the recruitment of immune cells to the arterial wall and atherosclerotic lesion⁴⁸, they also play a pivotal role in atherothrombosis: intravascular damage caused by the rupture of an atherosclerotic plaque, which leads to massive occlusive platelet aggregation thereby causing tissue ischemia, clinically manifested as myocardial infarction or stroke⁴⁹. Platelet CD40L has been reported to be involved in several immunological aspects of platelet function. In platelets, CD40L can signal via its receptor CD40, and activate NF κ B signaling via TRAF2, thereby contributing to platelet activation and release of inflammatory chemokines⁵⁰. Platelet CD40L facilitates leukocyte recruitment to the arterial wall, the formation of platelet-leukocyte aggregates and platelet chemokine secretion²⁷, all mechanisms related to atherosclerosis development. However, the CD40-TRAF2 pathway only reflects a minor part of the CD40L cascade in platelets, and the interaction between CD40L and the integrin α IIb β 3 appeared to be more relevant^{26,51}. Deficiency of CD40L- α IIb β 3 interactions resulted in less α IIb β 3 activation, decreased thrombus growth, and increased thrombus stability, as also observed in the platelet-specific (Cd40^{fl/fl}/Pf4Cre) model in our study, which appeared independent of the receptor CD40⁵¹. In the present study, we did not observe any effect of selective deficiency of platelet CD40L on atherosclerotic plaque burden or phenotype. However, we did find a beneficial role for deficiency of platelet CD40L in an atherothrombosis model, confirming the importance

of platelet CD40L in thrombus formation and stability. At first sight these findings seem contradictory to a previous study, where we demonstrated that repeated injection of thrombin activated CD40L-deficient platelets reduced atherosclerosis when compared to injection of thrombin activated wild type platelets by decreasing leukocyte recruitment, PLAs and thrombus formation²⁷. However, these results actually show that platelet CD40L affects atherogenesis in models causing a certain level of platelet activation.'

The authors speculate on cell selective inhibition of CD40L. Are there convincing therapeutic concepts in this regards? The authors should at least mention these.

We agree with reviewer #4, that cell selective inhibition of CD40L is still speculative. However, recent developments, including bi-specific antibodies (i.e. CD3-CD40L), as is used in cancer therapy, have opened the possibility to specifically target T cells in immunotherapeutic approaches. We have integrated this into our discussion (revised manuscript, pages 12-13, Lines 266-270):

'By deciphering the cell type-specific effects of the CD40L-CD40 pathway in disease, targeted therapeutic strategies directed against cell types (i.e., via cell-specific bi-specific antibodies⁵⁵ or targeted nanobiologics⁵⁶) or cell type-associated signaling intermediates, we will be able to more precisely target the desired effector pathways of the CD40L-CD40 dyad, thereby limiting unwanted side effects.'

The authors cite literature that CD40L is a risk marker for various diseases, including cardiovascular events. However, it seems the authors are quite biased in their selection of literature. There are also reports that CD40L is not suitable or a strong biomarker. The authors should discuss this in a more unbiased way.

We agree with the reviewer that there are also reports that sCD40L is not a strong biomarker. We have included these papers in the introduction.

The authors should discuss the potential interaction of CD40L with ligands of CD40L others than CD40.

We agree with reviewer #4: CD40L indeed can bind to other receptors than CD40, including Mac1 and $\alpha 5\beta 1$ integrin, as well as $\alpha 2\text{Ib}\beta 3$ integrin. We have now included this in the introduction section (revised manuscript, pages 4, Lines 29-31):

'Although CD40 is the most common receptor for CD40L in immune cells²³, CD40L can also bind to the integrin complexes $\alpha_M\beta 2$ ²⁴ and $\alpha 5\beta 1$ ²⁵. Platelet CD40L, on the other hand, mainly signals through the integrin $\alpha\text{IIb}\beta 3$ leading to platelet aggregation and endothelial activation^{26,27}

Figure 1: The scope of investigation of atherosclerosis is quite limited. Collagen content, necrotic core size, abundance of various leukocyte subsets and other measures normally assessed for atherosclerosis would be of interest. The authors should also explain why they used chow diet and not high fat diet to investigate atherosclerosis in the ApoE^{-/-} mouse model.

We agree with the reviewer that our initial plaque characterization was limited. Therefore, we have further characterized our T cell CD40L model in Figure 1. First, we have included an analysis of major immune cell populations in the blood (Supplemental Figure 4a). Next, we have described changes in aortic cytokines with particular focus on the decrease in IL-6 and IFN γ (Figure 1h). Finally, we have added analyses of collagen content, necrotic core content, as well as the Virmani classification, and we found that T-cell deficient CD40L mice not only have smaller plaques, but that these plaques also contain less T cells, less macrophages, smaller necrotic cores, more VSMCs and thicker fibrous caps.

With regards to the animal model, our goal was to study late-stage atherosclerosis. ApoE mice are unique in comparison to LDLr deficient in that they do not require a high fat diet to induce atherosclerosis. Therefore, long term chow diet is a commonly used method to study late-stage atherosclerosis. Generally, this model removes the variable of diet-based hyper-inflammation that may cause biological variance between animals.

Figure 2: These results are not really novel. The authors should clearly state this. Can the authors provide platelet staining in the injured arteries? Can the authors show that CD40L deficiency in T cells does not influence atherothrombosis?

We respectfully disagree that these results are not novel. To our knowledge, this is the first study showing a wire injury model in a platelet-specific CD40L-deficient atherosclerotic mouse model. We are aware of wire injury models in the CD40L^{-/-} mouse, and the protective effects of CD40L-deficiency, but in that study, besides absence of platelet CD40L, the absence of endothelial cell CD40L, as well as VSMC CD40L and T cell CD40L all affect the phenotypic outcome (PMID: 18349125).

We cannot rule out that T cell CD40L deficiency does not affect atherothrombosis, as T cells are also involved in wound healing. However, from a platelet perspective, our T-cell deficient CD40L deficient mice do not exhibit extended bleeding or clotting times (data not shown).

Figure 6d: The stainings are difficult to assess. Can the authors provide a quantification? How many experiments are done here? IgM levels have been described as being protective in atherosclerosis. How does that fit to the authors' results? Also, the literature on anti-oxLDL IgG is somehow controversial which needs further discussions.

We agree that the stainings are difficult to interpret and do not add to the paper. Therefore, we have removed these stainings from this figure (Supplemental Figure 6).

We also discussed the potential impact of the decreased IgG and anti-oxLDL IgG levels observed in our CD4-CD40L-ApoE^{-/-} mice in the revised version of the manuscript (page 11, Lines 224-233):

'Another phenotypic aspect of our model that, besides a decreased Th1 response, will have contributed to the observed decrease in atherosclerosis, is the reduction in IgG, and more specifically anti-oxLDL IgG antibodies. Germinal center derived IgG antibodies are known to promote atherogenesis and do this via multiple atherogenic pathways including metabolic reprogramming of immune cells, growth factors as well as immune cell recruitment⁴⁴. In addition, antibodies against oxidation specific epitopes, including oxidized (ox)LDL also play a major role in atherosclerosis⁴⁵. Whereas natural IgM oxLDL antibodies are considered to protect against atherosclerosis⁴⁶, via the induction of IL5, IgG antibodies against oxLDL are considered pro-atherogenic⁴⁷. The lack of Ig isotype switching, characteristic of CD40(L) deficiency, was also observed in our T cell deficient CD40L mice where total IgG, as well as oxLDL specific IgG, had decreased thereby contributing to the observed decrease in atherosclerosis.'

REVIEWERS' COMMENTS

Reviewer #1 (Remarks to the Author):

The authors have done an excellent job in responding to and addressing my concerns. I feel the revised manuscript is considerably improved. I have no further substantive concerns about this report.

Reviewer #2 (Remarks to the Author):

I feel the authors have made a good-faith effort to provide adequate responses to the many critiques of the reviewers. The attempt to explain the many complex roles of CD40L/CD40 is daunting and the many revisions are helpful.

The new human data correlating CD40 indices with INF γ are interesting but the various correlations ($r = .18-.26$) are so weak that I am not sure these add much quantitative support other than a very minor role--you may want to rethink if these are worth reporting.

Reviewer #4 (Remarks to the Author):

The authors have addressed my previous comments adequately and have clearly invested major efforts to improve their manuscript.

Reviewer #5 (Remarks to the Author):

It is a well done revision.

1. Authors clarified all reviewer's concerns and provide additional data.
2. They removed CHS data that did not reflect the events during atherogenesis according to reviewer's suggestion.
3. Additionally, authors moved data presented in Fig. 6 to Suppl. Figures as the data is not in the central point of their manuscript.

I have some minor points.

1. Discussion, line 223: Authors say "...likely partly dysfunctional, Tregs"

While Fig. 3d shows reduced percentage of Tregs in blood, lymph node and spleen, the data presented in Fig. 3e does not seem to show dysfunction of Tregs.

Maybe authors could consider rephrasing the sentence or explain it.

2. Data

Data presented in suppl. Fig. 6d show reduced IgG2a in CD4Cre-CD40L^{fl/fl}-ApoE⁻ mice what can be caused not only by lack in CD40-CD40L isotype switching. Authors already showed that this defective strain of mice has also reduced percentage of Th1 cells (Fig. 3b) and produce less IFN- γ what can additionally affect IgG production. Maybe this should be mentioned in the manuscript.

line 203: "that" is duplicated

line 207: should be IL-2 not IL2

line 230: should be IL-5 not IL5

Response to reviewers:

Reviewer #1

The authors have done an excellent job in responding to and addressing my concerns. I feel the revised manuscript is considerably improved. I have no further substantive concerns about this report.

We thank reviewer 1 for his/her positive comments.

Reviewer #2

I feel the authors have made a good-faith effort to provide adequate responses to the many critiques of the reviewers. The attempt to explain the many complex roles of CD40L/CD40 is daunting and the many revisions are helpful.

We thank reviewer 2 for his/her positive comments.

The new human data correlating CD40 indices with INF γ are interesting but the various correlations ($r = .18-.26$) are so weak that I am not sure these add much quantitative support other than a very minor role--you may want to rethink if these are worth reporting.

We agree with reviewer 2 that the data are interesting. Although the correlations are weak, they are significant and we feel that these data are worthwhile to be reported and important to know for the vascular community to spark further research regarding this correlations.

Reviewer #4

The authors have addressed my previous comments adequately and have clearly invested major efforts to improve their manuscript.

We thank reviewer 4 for his/her positive comments.

Reviewer #5

It is a well done revision.

1. Authors clarified all reviewer's concerns and provide additional data.
2. They removed CHS data that did not reflect the events during atherogenesis according to reviewer's suggestion.
3. Additionally, authors moved data presented in Fig. 6 to Suppl. Figures as the data is not in the central point of their manuscript.

We thank reviewer 5 for his/her suggestions in the previous review round and his/her compliment.

I have some minor points.

1. Discussion, line 223: Authors say ".....likely partly dysfunctional, Tregs"

While Fig. 3d shows reduced percentage of Tregs in blood, lymph node and spleen, the data

presented in Fig. 3e does not seem to show dysfunction of Tregs.
Maybe authors could consider rephrasing the sentence or explain it.

We agree with reviewer 5 that our CD40L^{-/-} Tregs are not more or less suppressive than their wild type counterparts. However, 1 suppression assay in vitro does not completely rule out a potential dysfunction in vivo. We have therefore rephrased the sentence:

“As we did not see alterations in the suppressive capacities of CD40L-deficient Tregs decrease in vitro, we assume that the decrease in Th1 response in our T cell-specific CD40L (*Cd40^{fl/fl}/Cd4Cre^{tg}*) deficient mice with advanced atherosclerosis affects atherogenesis to a larger extent than the decrease in, potentially partly dysfunctional, Tregs.”

2. Data

Data presented in suppl. Fig. 6d show reduced IgG2a in CD4Cre-CD40L^{fl/fl}-ApoE⁻ mice what can be caused not only by lack in CD40-CD40L isotype switching. Authors already showed that this defective strain of mice has also reduced percentage of Th1 cells (Fig. 3b) and produce less IFN- γ what can additionally affect IgG production. Maybe this should be mentioned in the manuscript.

We thank reviewer 5 for pointing this out and we have now mentioned the effects on the reduced Th1 and IFN γ responses on IgG production:

“Another phenotypic aspect of our model that, besides, or even due to a decreased Th1 response, will have contributed to the observed decrease in atherosclerosis, is the reduction in IgG, “

line 203: "that" is duplicated
We have corrected this typo.

line 207: should be IL-2 not IL2
We have corrected this typo.

line 230: should be IL-5 not IL5
We have corrected this typo.